# The HP1 box of KAP1 organizes HP1α for silencing of endogenous retroviral elements in embryonic stem cells

Nitika Gaurav [1,11], Ryan O'Hara [2,11], Usman Hyder [3], Weihua Qin[4], Cheenou Her[5], Hector Romero [6], Amarjeet Kumar [7], Maria J. Marcaida [8], Rohit K. Singh[1], Ruud Hovius[9], Karthik Selvam [1], Jiuyang Liu[1], Sara Martire [2], Yuhang Yao [2], Ashwini Challa[3], Matteo Dal Peraro [8], Beat Fierz [9], Hidetoshi Kono [7,10], M. Cristina Cardoso [6], Galia T. Debelouchina [5], Heinrich Leonhardt[4], Iván D'Orso [3] ✉, Laura A. Banaszynski [2] ✉ & Tatiana G. Kutateladze [1] ✉

Repression of endogenous retroviral elements (ERVs) is facilitated by KAP1 (KRAB-associated protein 1)-containing complexes, however the underlying mechanism remains unclear. Here, we show that binding of KAP1 to the major component of the heterochromatin spreading and maintenance network, HP1α, plays a critical role in silencing of repetitive elements. Structural, biochemical and mutagenesis studies demonstrate that the association of the HP1 box of KAP1 (KAP1_{Hbox}) with the chromoshadow domain of HP1α (HP1α_{CSD}) leads to a symmetrical arrangement of HP1α_{CSD} and multimerization that may promote the closed state of chromatin. The formation of the KAP1_{Hbox}-HP1α_{CSD} complex enhances charge driven DNA binding and phase separation activities of HP1α. ChIP-seq and ATAC-seq analyses using KAP1 knock out mouse embryonic stem cells expressing wild type KAP1 or HP1-deficient KAP1 mutant show that in vivo, KAP1 engagement with HP1 is required for maintaining inaccessible chromatin at ERVs. Our findings provide mechanistic and functional insights that further our understanding of how ERVs are silenced.

The formation and maintenance of heterochromatin, a highly condensed and transcriptionally inactive state of chromatin, is vital for genome stability[1]. Heterochromatin is enriched in repetitive elements, including retrotransposon sequences and long and short interspersed elements that must be repressed to sustain normal cell developmental programs and prevent chromosomal instability[2]. Failure to silence repetitive elements is linked to cancer, neurodegenerative diseases, and aging[3,4]. One of the major mechanisms that mediates this silencing and heterochromatin compaction in mammals depends on activities of the heterochromatin protein 1 paralogues (HP1α, HP1β and HP1γ)[5].

[1]Department of Pharmacology, University of Colorado School of Medicine, Aurora, CO, USA. [2]Cecil H. and Ida Green Center for Reproductive Biology Sciences, The University of Texas Southwestern Medical Center, Dallas, TX, USA. [3]Department of Microbiology, The University of Texas Southwestern Medical Center, Dallas, TX, USA. [4]Department of Biology II, Ludwig-Maximilians-Universität München, München, Germany. [5]Department of Chemistry and Biochemistry, University of California San Diego, La Jolla, CA, USA. [6]Department of Biology, Technical University of Darmstadt, Darmstadt, Germany. [7]Institute for Quantum Life Science, National Institutes for Quantum Science and Technology, Chiba, Japan. [8]Institute of Bioengineering, School of Life Sciences, Ecole Polytechnique Fédérale de Lausanne, Lausanne, Switzerland. [9]Institute of Chemical Sciences and Engineering, Ecole Polytechnique Fédérale de Lausanne, Lausanne, Switzerland. [10]Graduate School of Science and Engineering, Chiba University, Chiba, Japan. [11]These authors contributed equally: Nitika Gaurav, Ryan O'Hara. ✉e-mail: Ivan.Dorso@UTSouthwestern.edu; laura.banaszynski@utsouthwestern.edu; tatiana.kutateladze@cuanschutz.edu

HP1 proteins play both recruitment roles in heterochromatin assembly and structural roles in higher order heterochromatin formation[6]. Understanding the molecular principles governing these functions is essential to our understanding the deleterious effects of loss of heterochromatin in development and disease.

HP1α contains two folded domains, chromodomain (CD) and chromoshadow domain (CSD), which are linked by a hinge region and flanked by the amino-terminal and carboxyl-terminal extensions (nte and cte). CD binds to tri/di methylated lysine 9 of histone H3 (H3K9me3/2), a major epigenetic modification and a hallmark of heterochromatin, whereas CSD mediates dimerization and associates with various proteins through recognizing the PxVxL motif, also known as the HP1 box[7–12]. The hinge region has DNA and RNA binding activity and along with nte and cte is subjected to post translational modifications (PTMs)[13]. Several PTMs have been shown to affect the interactions of HP1α and thus alter its functions, e.g. phosphorylation was found to promote a liquid-liquid phase separation capability of HP1α[14,15].

HP1α binds to the HP1 box of the methyltransferase SUV39H1 that generates H3K9me3 forming new binding sites for HP1α and therefore provides a plausible model for heterochromatin spreading and maintenance. However, a relatively weak binding of HP1α to H3K9me3 suggests that interactions with other heterochromatin co-factors, including co-repressors, such as the KRAB-associated protein 1 (KAP1), also known as tripartite motif-containing protein 28 (Trim28) and TIF1β, could be essential in this model. KAP1 cooperates with the histone deacetylase NuRD and another H3K9-specific methyltransferase SETDB1 in promoting chromatin condensation and transcriptional repression and plays a major role in silencing of endogenous retroviral elements (ERVs)[16–20]. This silencing is initiated through SMARCAD1-dependent nucleosome turnover resulting in histone variant H3.3 deposition at these regions[21,22], which stabilizes KAP1/SETDB1 at ERVs through interactions with its chaperone proteins ATRX and DAXX[23–25]. In addition to supporting the establishment of H3K9me3 as a recruitment module for HP1, KAP1 contains a HP1 box (KAP1_Hbox) sequence, which is recognized by the CSD of HP1α (HP1α_CSD) in biochemical studies[26]. The importance of the direct KAP1-HP1 interaction in ERV silencing has not been established.

In this work, we highlight the critical role of HP1α and KAP1 in silencing of repetitive elements and define the molecular mechanism underlying their cooperative action. We show that binding of KAP1_Hbox leads to the structural organization and multimerization of HP1α_CSD, enhances DNA binding and phase separation activities of HP1α, and that HP1 binding function of KAP1 is required to maintain inaccessible chromatin at ERVs in mouse embryonic stem cells (ESCs).

## Results and discussion

### Colocalization of HP1 and KAP1 at ERVs in ES cells

The findings that KAP1-containing complexes mediate transcriptional repression of ERVs and KAP1 binds to HP1 proteins[16,26] prompted us to investigate the mechanisms by which KAP1 and HP1 cooperate in mediating ERVs silencing. To explore this relationship, we analyzed chromatin immunoprecipitation coupled with deep sequencing (ChIP-seq) datasets previously generated by us and others[23,27], as well as newly acquired data from mouse ESCs, and assessed chromatin occupancies of HP1 proteins genome wide. As shown in Fig. 1a–f, enrichment of HP1 proteins correlated with each other and with KAP1 enrichment. To identify repeat families associated with HP1 enrichment, we mapped HP1 data sets to a comprehensive database of repetitive sequences in the mouse genome. Unbiased hierarchical clustering demonstrated a clear enrichment of HP1 proteins over a subset of ERVs, as well as enrichment of KAP1 and other elements of the retroelement silencing machinery, including H3K9me3 and H3.3 (Fig. 1g and Supplementary Fig. 1). The most evident enrichment of HP1 proteins was observed at intracisternal A-type particles (IAPs), MERVK10C and early transposon (ETn) ERVs and the 5'UTR of intact

and evolutionarily young LINE elements, in agreement with previous studies[21,28]. While both ERVs and LINEs showed enrichment of H3K9me3 and HP1 proteins, LINEs showed strong enrichment of transcription-associated histone modifications such as H3K9ac and H3K27ac, whereas ERVs were depleted of these modifications (Fig. 1g and Supplementary Fig. 1), suggesting distinct molecular regulation of these elements in ESCs. All retroelements were depleted of H3K27me3, a hallmark of the polycomb PRC2 complex activity. Co-enrichment of HP1s and KAP1 was observed both at the level of individual IAPEz, MERVK10C and ETn elements (Fig. 1h) and genome-wide (Fig. 1i–k and Supplementary Fig. 2). Overall, substantial overlap in occupancies of HP1s and KAP1 at retroelements suggested a cooperative function in ESCs.

### Molecular basis of the KAP1_Hbox recognition

The commonly accepted mechanism for HP1s and KAP1 colocalization involves SETDB1 mediated deposition of H3K9me3 supported by KAP1 and recognized by HP1, and interaction between KAP1 and HP1 has also been reported[6,19,20,26,29]. Although the direct association of KAP1 with HP1α_CSD was identified in 2000[26], mechanistic details of this association remain undetermined. We used crystallographic, NMR, fluorescence and immunoprecipitation approaches to define the molecular basis for binding of KAP1 to HP1α_CSD (Fig. 2). The formation of a tight HP1α_CSD-KAP1 complex was confirmed by measuring binding affinity of HP1α_CSD to the HP1 box-containing KAP1 (KAP1_Hbox) peptide (aa 483–493 of KAP1) using fluorescence spectroscopy (Fig. 2b). To characterize the binding mechanism at atomic-resolution level, we co-expressed HP1α_CSD and a larger fragment of KAP1 (aa 468–496, KAP1_Hbox'), purified the HP1α_CSD-KAP1_Hbox' complex, crystallized it, and determined its structure by X-ray crystallography (Fig. 2c–f and Supplementary Table 1).

In the crystal structure, HP1α_CSD forms a homodimer (HP1α_CSD1 and HP1α_CSD2, colored pink and grey, respectively), also observed in the structures of CSDs of HP1β and HP1γ[10,11]. Each HP1α_CSD protomer folds into a short N-terminal helix α1, a central three-stranded antiparallel β-sheet, and two C-terminal helices α2 and α3 (Fig. 2c). The α3 helices of both HP1α_CSD protomers form the dimerization interface, packing against each other in a parallel manner (Fig. 2c–e). The KAP1_Hbox' is bound in an extended conformation in an elongated groove created by the C-terminal regions following α3 of HP1α_CSD. These regions of HP1α_CSD adopt a β-strand conformation and sandwich KAP1_Hbox', essentially forming a triple stranded β-sheet, with KAP1_Hbox' being oriented parallel to the HP1α_CSD1 region and antiparallel to the HP1α_CSD2 region (Fig. 2c).

A set of hydrophobic interactions and hydrogen bonds stabilize the HP1α_CSD-KAP1_Hbox complex. The side chains of Pro486, Val488 and Leu490 of KAP1_Hbox' are oriented toward and buried in the hydrophobic groove of the HP1α_CSD dimer. Characteristic intermolecular β-sheet hydrogen bonds are formed between the backbone amides of Arg487 and Ser489 of KAP1_Hbox' and backbone carbonyl groups of Thr173 and Arg171 of HP1α_CSD2. The backbone-backbone contacts are also observed between oxygen atoms of the carbonyl groups of Val488 and Leu490 of KAP1_Hbox' and the amides of Thr173 and His175 of HP1α_CSD1, as well as between amide of Leu490 of KAP1_Hbox' and the carbonyl of Thr173 of HP1α_CSD1. Additionally, the side chain hydroxyl group of Thr173 of HP1α_CSD1 forms a hydrogen bond with the backbone carbonyl group of Val488 of KAP1_Hbox'.

### Mapping the HP1α_CSD-KAP1_Hbox binding interface

To characterize the formation of the HP1α_CSD-KAP1_Hbox complex in solution, we carried out NMR experiments. We generated $^{15}$N-labeled HP1α_CSD and recorded its $^{1}$H,$^{15}$N heteronuclear single quantum coherence (HSQC) spectra as unlabeled KAP1_Hbox was titrated into the NMR sample (Fig. 3a and Supplementary Fig. 3). Large chemical shift perturbations (CSPs) were induced in HP1α_CSD by addition of KAP1_Hbox.

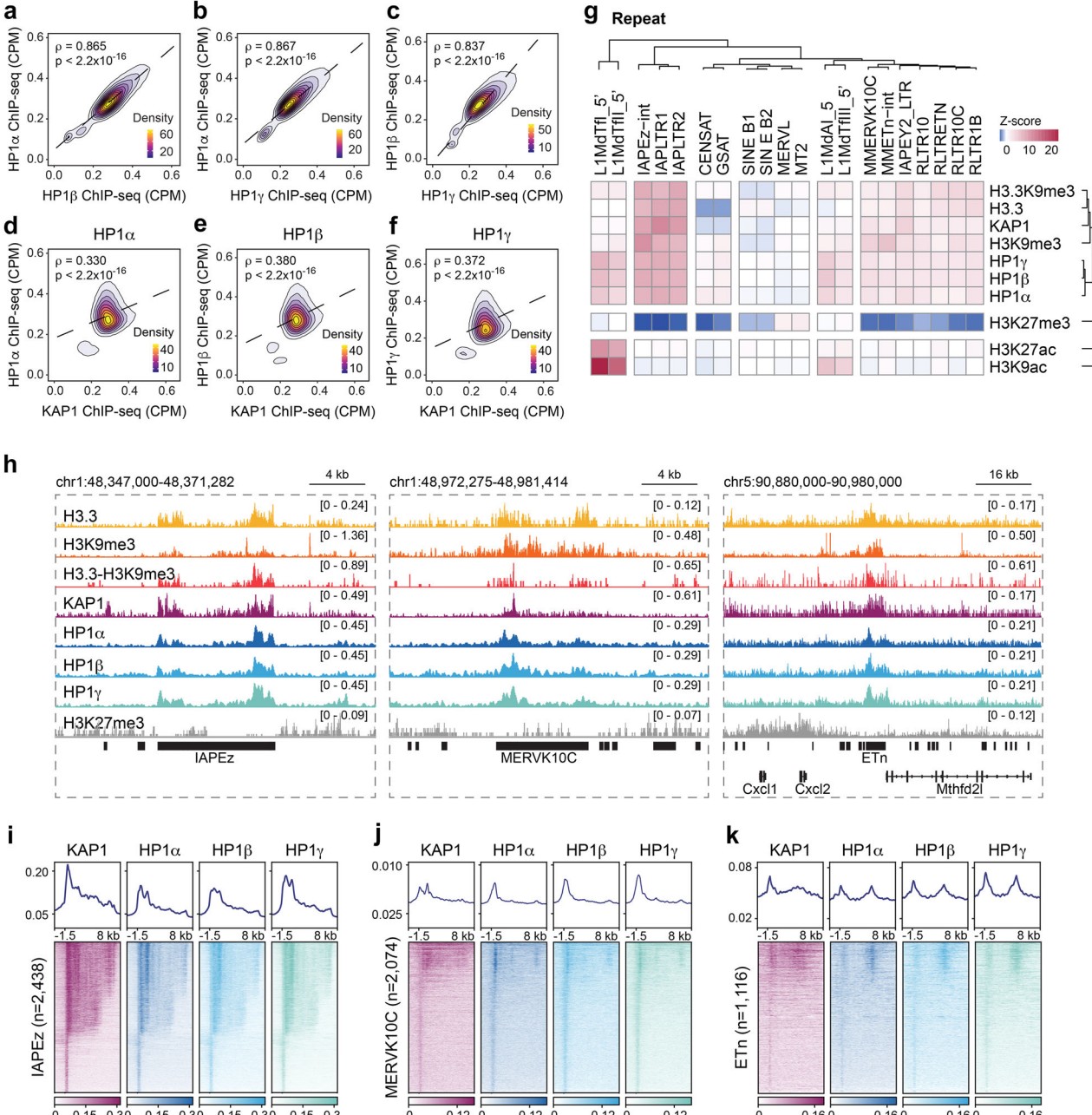

**Fig. 1 | KAP1 and HP1 are enriched at endogenous retroelements in mouse embryonic stem cells. a–c** Correlation plots between HP1 proteins genome-wide at 5 kb bins in ESCs. Spearman correlation tests were used to determine statistical significance. Correlation plots between KAP1 and (**d**) HP1α, (**e**) HP1β and (**f**) HP1γ genome-wide at 5 kb bins in ESCs. Spearman correlation tests were used to determine statistical significance. **g** ChIP-seq enrichment of heterochromatic histone modifications and factors, including HP1 proteins, mapped to the repetitive genome. Data are represented in a hierarchically (Spearman rank) clustered heatmap of z-score fold enrichment (red) or depletion (blue) over a matched input. See Supplementary Fig. 1 for complete heatmap. **h** Genome browser representations of enrichment of heterochromatic histone modifications and factors, including HP1 proteins, in ESCs at IAPEz (left), MERVK10C (center) and ETn (right). The y-axis represents read density in counts per million mapped reads (CPM). Average profiles (top) and heatmaps (bottom) of KAP1, HP1α, HP1β and HP1γ enrichment at IAPEz (*n* = 2,438), (**j**) MERVK10C (*n* = 2,074) and (**k**) ETn (*n* = 1,116) in ESCs. Data are centered on the LTR with 1.5 kb upstream and 8 kb downstream of the LTR displayed for each analysis.

Notably, a set of resonances corresponding to the unbound state of HP1α_CSD decreased in intensity and disappeared, and concomitantly, another set of resonances that corresponds to the KAP1_Hbox-bound state of HP1α_CSD gradually appeared. This pattern of CSPs is characteristic of the slow exchange regime on the NMR time scale and indicates tight binding.

To identify the residues of HP1α_CSD involved in contact with KAP1_Hbox, we performed triple resonance NMR experiments on uniformly $^{13}$C,$^{15}$N-labeled HP1α_CSD' and assigned backbone amide

resonances (Supplementary Fig. 4). To define the binding interface, CSPs observed in $^{1}$H,$^{15}$N HSQC spectra of HP1α_CSD upon addition of KAP1_Hbox were plotted per residue (Fig. 3b). As shown in Fig. 3b, two regions of HP1α_CSD, encompassing Gly128-Phe138 and Tyr168-Trp174, were particularly perturbed, with Gly128, Thr130, Asp131, Cys133, Gly134, Met137, Phe138, Tyr168, Glu169, Leu172 and Trp174 peaks disappearing at the HP1α_CSD:KAP1_Hbox molar ratio of 1:1. Mapping the most perturbed residues onto the structure of HP1α_CSD in complex with KAP1_Hbox' outlined the interface with KAP1_Hbox and indicated that

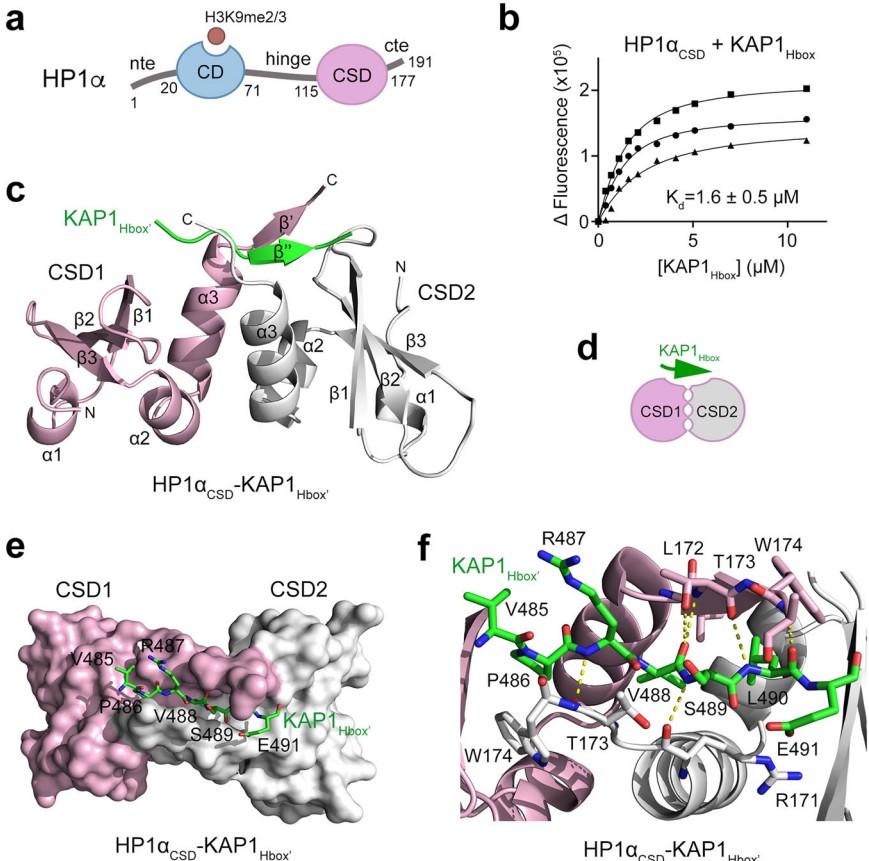

**Fig. 2 | Structural basis for the interaction of KAP1$_{Hbox}$ with HP1α$_{CSD}$. a** Domain architecture of HP1α: nte amino-terminal extension, CD chromodomain, CSD chromoshadow domain, cte carboxyl-terminal extension. Histone H3K9me2/3 mark recognized by CD is depicted as purple circle. **b** Binding curves used to determine binding affinity of HP1α$_{CSD}$ for the KAP1$_{Hbox}$ peptide by tryptophan fluorescence. $K_d$ is represented as average ±SD of three independent experiments. $n = 3$. **c** The crystal structure of the HP1α$_{CSD}$ dimer in complex with KAP1$_{Hbox'}$ is depicted as a ribbon diagram with one chromoshadow domain protomer labeled as CSD1 and colored pink and another protomer labeled as CSD2 and colored grey.

Bound KAP1$_{Hbox'}$ is shown as a green ribbon that pairs with the C-terminal parts of CSD1 in a parallel manner and CSD2 in an antiparallel manner (both are in an extended conformation). **d** A schematic representation of the HP1α$_{CSD}$ dimer in complex with KAP1$_{Hbox}$. **e** Surface representation of the HP1α$_{CSD}$-KAP1$_{Hbox'}$ complex colored as in (**c**). KAP1$_{Hbox'}$ is shown as green stick. Residues of KAP1$_{Hbox'}$ involved in the interaction with HP1α$_{CSD}$ are labeled. **f** Close-up view of the KAP1$_{Hbox}$-binding site of the HP1α$_{CSD}$ dimer. Residues involved in the interaction between HP1α$_{CSD}$ and KAP1$_{Hbox'}$ are labeled. Dashed lines represent hydrogen bonds. Source data are provided as a Source Data file.

the binding mode observed in the crystal structure of HP1α$_{CSD}$ is conserved in solution (Fig. 3b, c).

Mutations of the hydrophobic residues in the HP1α$_{CSD}$-KAP1$_{Hbox}$ interface, Trp174 of HP1α$_{CSD}$ or Val488 of KAP1$_{Hbox}$, abolished this interaction, as no CSP were observed in the respective $^1$H,$^{15}$N HSQC titration experiments (Fig. 3d, e). The HP1α$_{CSD}$-KAP1$_{Hbox}$ interaction was also disrupted when Ile165 in α3, which is required for the dimerization of HP1α, was substituted with a glutamate (Fig. 3f and Supplementary Fig. 5). In agreement, endogenous KAP1 was immunoprecipitated from HCT116 cells by full-length wild type HP1α but not by the W174A or I165E mutants of HP1α (Fig. 3g), and the V488E mutation in full-length KAP1 eliminated binding of exogenously expressed FLAG-tagged HP1α (Fig. 3h). Together, these data pointed to the importance of hydrophobic and van der Waals contacts in stabilization of the HP1α$_{CSD}$-KAP1$_{Hbox}$ complex and confirmed that this complex is also formed in context of the full-length proteins.

## KAP1$_{Hbox}$ enhances binding of HP1α to DNA

HP1α has been shown to bind DNA, compacting it into stable phase separated domains[13,30]. To assess the impact of the interaction with KAP1$_{Hbox}$, we examined the association of full-length (FL) HP1α with 147 bp Widom 601 DNA (DNA$_{147}$) in an electrophoretic mobility shift assay (EMSA) with and without KAP1$_{Hbox}$ and KAP1$_{Hbox'}$. Increasing

amounts of HP1α were incubated with DNA$_{147}$, and the reaction mixtures were resolved on a native polyacrylamide gel (Fig. 4a and Supplementary Fig. 6). A gradual increase in HP1α concentration resulted in the shift of the DNA band, indicating that HP1α forms a complex with DNA$_{147}$, and the presence of KAP1$_{Hbox}$ or KAP1$_{Hbox'}$ augmented binding of HP1α to DNA (Fig. 4a and Supplementary Fig. 6). As expected, the isolated HP1α$_{CSD}$ either in the presence or absence of KAP1$_{Hbox}$ was unable to shift the DNA band (Supplementary Fig. 7a), supporting previous reports that the DNA binding activity of HP1α depends on the hinge region linking HP1α$_{CD}$ and HP1α$_{CSD}$[13,30]. We note that the KAP1 peptide contains four positively charged residues located in (R487) and around (R483, K484 and R492) the binding interface and one negatively charged residue (E491), therefore, we hypothesized that three positive charges acquired due to the complex formation can alter electrostatic landscape of HP1α and lead to a tighter interaction with the negatively charged DNA. To test this idea, we performed EMSA assays of DNA$_{147}$ and HP1α in the presence of KAP1$_{Hbox}$ peptides in which R487 or three residues R483, K484 and R487 simultaneously were substituted with an aspartate. As shown in Fig. 4b, both modified KAP1$_{Hbox}$ peptides substantially decreased binding of HP1α to DNA, with the replacement of a single R487 residue impacting most. R487 is located in the middle of the β-strand in the HP1α$_{CSD}$-KAP1$_{Hbox'}$ complex and therefore is highly restrained with its positively charged side chain

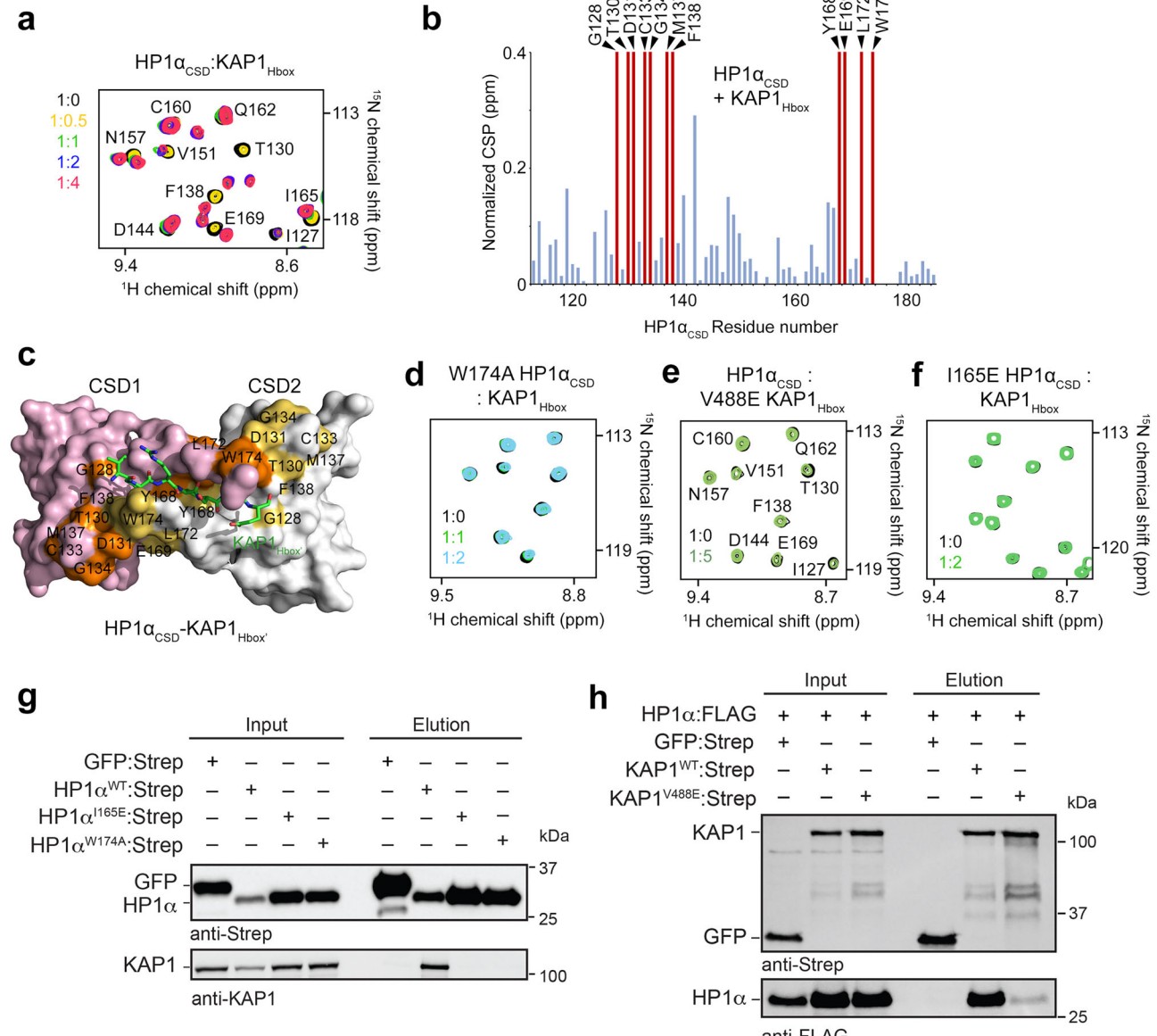

**Fig. 3 | Mapping the KAP1Hbox-HP1αCSD binding interface. a** Overlayed $^1$H,$^{15}$N HSQC spectra of $^{15}$N-labeled HP1α$_{CSD}$ recorded in the presence of increasing amount of KAP1$_{Hbox}$ peptide. Spectra are color coded according to the protein:peptide molar ratio. **b** Bar plot of normalized CSPs in $^1$H,$^{15}$N HSQC spectra of $^{15}$N-labeled HP1α$_{CSD}$ induced by the fourfold molar excess of KAP1$_{Hbox}$ peptide. Disappeared peaks are indicated by red bars and labeled. **c** Surface representation of the HP1α$_{CSD}$-KAP1$_{Hbox'}$ complex, colored as in Fig. 2c. KAP1$_{Hbox'}$ is shown as green stick. The residues of HP1α$_{CSD}$, resonances of which disappear in (**b**) are mapped onto the surface, colored orange and yellow and labeled. **d** Overlayed $^1$H,$^{15}$N HSQC spectra of the $^{15}$N-labeled W174A mutant of HP1α$_{CSD}$ recorded in the presence of

increasing amount of wild type KAP1$_{Hbox}$ peptide. **e** Overlayed $^1$H,$^{15}$N HSQC spectra of $^{15}$N-labeled HP1α$_{CSD}$ recorded in the presence of the V488E mutant of KAP1$_{Hbox}$ peptide. Spectra are color coded according to the protein:peptide molar ratio. **f** Overlayed $^1$H,$^{15}$N HSQC spectra of the $^{15}$N-labeled I165E mutant of HP1α$_{CSD}$ recorded in the presence of increasing amount of KAP1$_{Hbox}$ peptide and color coded according to the protein:peptide molar ratio. Western blot analysis of pull-down assays with Strep-tagged wild type and mutated HP1α (**g**) and Strep-tagged wild type and mutated KAP1 with FLAG-tagged HP1α constructs (**h**). controls: Strep-tagged GFP. The data are representative of 3 biological replicates with identical results. $n = 3$ Source data are provided as a Source Data file.

being fully accessible for DNA binding (Fig. 2e, f). Collectively, these results revealed that the association with KAP1$_{Hbox}$ enhances binding of HP1α to DNA, and R487 of KAP1 is essential. Furthermore, the increase in DNA binding activity was observed only upon formation of the HP1α$_{CSD}$-KAP1$_{Hbox}$ complex and KAP1$_{Hbox}$ or KAP1$_{Hbox'}$ without HP1α did not appreciably bind DNA (Supplementary Fig. 7b) in agreement with previous reports showing that while KAP1 itself binds DNA, its HP1 box region is not involved[31,32].

### KAP1$_{Hbox}$ enhances phase separation capability of HP1α

Biochemical studies have demonstrated that HP1α undergoes liquid-liquid phase separation (LLPS) in vitro and in part sequesters

heterochromatin, forming phase separated compartments or puncta in cells[14,15,30,33-35], therefore we examined the role of KAP1 binding on the phase separation ability of HP1α. We generated KAP1$^{-/-}$ mouse embryonic stem cells (ESCs) stably expressing a GFP labeled KAP1 fragment (aa 114–834 of KAP1, GFP-KAP1$_{FR}$) or the P486A/V488A/L490A mutant of GFP-KAP1$_{FR}$ impaired in HP1α binding (GFP-KAP1$_{FR}$-PVL$_{mut}$) and monitored distribution of endogenous HP1α in these cells by immunofluorescence (Fig. 4c). Endogenous HP1α formed large puncta and colocalized with GFP-KAP1$_{FR}$ in KAP1$^{-/-}$ ESCs expressing wild type GFP-KAP1$_{FR}$. However, in KAP1$^{-/-}$ ESCs expressing GFP-KAP1$_{FR}$-PVL$_{mut}$, which showed a uniform distribution throughout the nucleus, endogenous HP1α was also more dispersed and formed

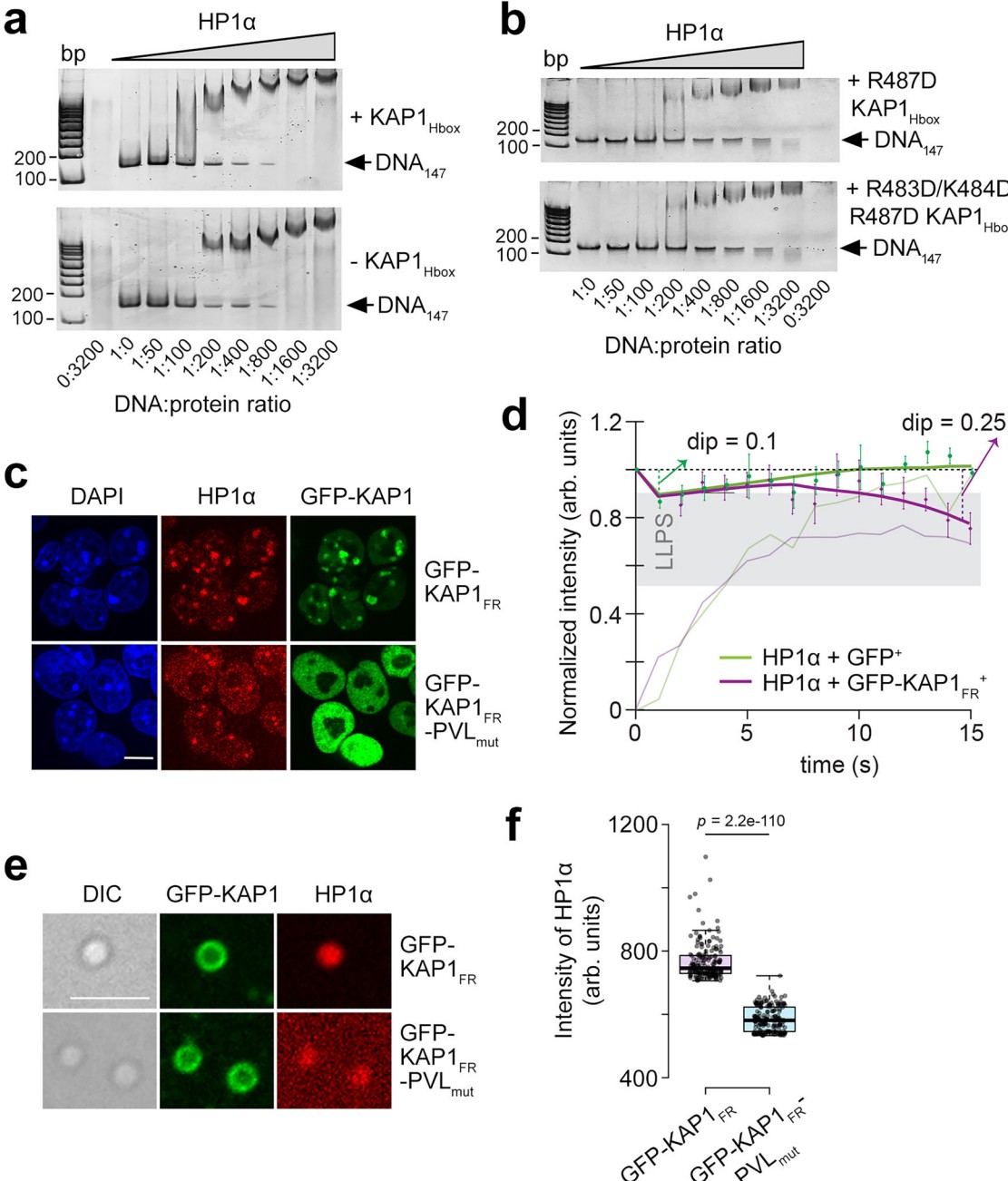

**Fig. 4 | KAP1Hbox enhances the phase separation and DNA binding abilities of HP1α. a** EMSA of 147 bp Widom 601 DNA (DNA_{147}) in the presence of increasing amounts of HP1α with (top) and without (bottom) tenfold molar excess of KAP1_{Hbox}. DNA:protein ratio is shown below the gel images. **b** EMSA of DNA_{147} in the presence of increasing amounts of HP1α with tenfold molar excess of indicated modified KAP1_{Hbox}. **c** Immunofluorescence images show the cellular distribution of GFP-KAP1 and endogenous HP1α in KAP1^{−/−} ESCs stably expressing GFP-KAP1_{RF} or GFP-KAP1_{FR}-PVL_{mut} stained with an anti-HP1α antibody (red). DNA was counterstained using DAPI (blue). Scale bar: 10 μm. **d** Half-FRAP recovery curves for HP1α in GFP^{+} (green) or GFP-KAP1_{FR}^{+} (purple) ESCs for 8–12 heterochromatin compartments. The bleach half recovery is indicated by thin lines, and a Savitzky-Golay fit was performed to show the recovery of the non-bleached half and calculate the dip

(thick lines)[36]. The non-bleached data points represent average ±SEM between 8 and 12 measurements. *n* > 8 The gray area indicates the range of dip depths in the non-bleached half that correspond to LLPS[36]. **e** Representative confocal images of DNA-induced HP1α phase separated droplets in the presence of GFP-KAP1_{FR} wild type (green) or GFP-KAP1_{FR}-PVL_{mut} (green) and 647 N labeled HP1α (red). DIC, Differential interference contrast. Scale bar: 5 μm. **f** The box plot displays the fluorescence of 647 N labeled HP1α within phase-separated droplets. Center lines show the medians; box limits indicate the 25th and 75th percentiles as determined by R software; whiskers extend 1.5 times the interquartile range from the 25th and 75th percentiles. *n* = 169 (wild type GFP-KAP1_{FR}), 175 (GFP-KAP1_{FR}-PVL_{mut}). Statistical analysis was performed by a two-sided Student's *t* test, *p*-value is indicated. Source data are provided as a Source Data file.

smaller and fewer puncta. These results indicated that the interaction with KAP1 promotes the formation of HP1α puncta in vivo.

To assess the effect of KAP1 binding in living cells, we performed fluorescence recovery after photobleaching (FRAP) experiments. We co-transfected ESCs with plasmids expressing dsRed-HP1α and either

GFP only or GFP-KAP1_{FR}, selected GFP^{+} cells and bleached half of heterochromatin compartments in these cells while measuring the recovery of fluorescence in both bleached and non-bleach heterochromatin regions (Fig. 4d). In the presence of GFP-KAP1_{FR}, neither the bleached half (thin purple line) nor the non-bleached half (thick purple

line) fully recovered their fluorescence, with the non-bleach half showing dips at 1 s and 15 s, indicative of the liquid-liquid phase separation property[36]. The large dip depth (0.25) in the HP1α intensity at 15 s indicated that the interior of this non-bleach heterochromatin compartment has a great degree of free diffusion, a sign of LLPS. In contrast, in the absence of GFP-KAP1$_{FR}$, HP1α fluorescence fully recovered in non-bleached half (thick green line) and almost fully recovered in the bleached half (thin green line). Together, these data suggested that KAP1$_{FR}$ enhances the LLPS ability of HP1α in living ESCs.

We then generated phase separated droplets in vitro by mixing HP1α and Widom 601 DNA and added GFP-KAP1$_{FR}$ and 647N labeled HP1α to the suspension, visualizing the droplets by confocal microscopy (Fig. 4e). As expected, either wild type GFP-KAP1$_{FR}$ or GFP-KAP1$_{FR}$-PVL$_{mut}$ impaired in HP1α binding concentrated at the periphery of the droplets due to the DNA binding activity of KAP1[31,32]. While GFP-KAP1$_{FR}$-PVL$_{mut}$ was unable to sequester 647N labeled HP1α into the condensed phase from non-condensed phase, wild type GFP-KAP1$_{FR}$ sequestered HP1a into the interior of droplets within a minute (Fig. 4e, f and Supplementary Fig. 8).

## HP1α oligomerizes through HP1α$_{CSD}$

To determine whether conformational changes accompany the complex formation, we obtained the crystal structure of the apo-state of HP1α$_{CSD}$ and characterized $^{15}$N-labeled full-length HP1α by NMR. Unexpectedly, we found that while resonances of HP1α$_{CD}$ were observed in the $^{1}$H,$^{15}$N HSQC spectrum of FL HP1α, most of the signals of HP1α$_{CSD}$ were undetectable, even when the spectra were collected on a 900 MHz NMR spectrometer (Fig. 5a, b). These data suggested that HP1α$_{CSD}$ may form oligomers larger than a dimer – HP1α$_{CSD}$ is a small globular module that should be amenable to NMR studies in the dimeric form – and corroborated previous analytical ultracentrifugation studies describing HP1α$_{CSD}$ dimer-tetramer equilibrium[26]. In support, mass photometry (MP) measurements showed that FL HP1α formed oligomers larger than a dimer of 49 kDa and oligomerization was concentration dependent (Fig. 5c). Titration of unlabeled HP1α$_{CSD}$ to the NMR sample of $^{15}$N-labeled I165E HP1α$_{CSD}$, which is incapable of dimerization through α3-helices, led to CSPs, indicating that the binding interface for this CSD-CSD interaction is different from the α-helix interface (Fig. 5d).

In agreement, the crystal structure of HP1α$_{CSD}$ in the apo-state showed that the HP1α$_{CSD}$ dimer is in contact with another HP1α$_{CSD}$ dimer (two dimers form an asymmetric unit, Supplementary Fig. 9a). The dimers interacted through their β-sheets (Fig. 5e, protomers of one HP1α$_{CSD}$ dimer are colored wheat and grey, and both protomers of the second dimer are colored yellow). The β-sheet binding interface was also identified in solution and mapped based on resonance perturbations observed in $^{15}$N-labeled HP1α$_{CSD'}$ upon titration with unlabeled HP1α$_{CSD'}$ (Fig. 5g, h and Supplementary Fig. 10).

The β-sheet binding interface was comprised of primarily hydrophobic residues, such as I126, I127, A129, L136, L139, W142, A148, L150 and V151 (Fig. 5f–h). To determine the role of the β-sheet interface in oligomerization of HP1α, we mutated L139 and L150, identified in molecular dynamics (MD) simulations, to glutamate in HP1α$_{CSD}$ and in FL protein and tested the L139E/L150E mutants by NMR and MP (Fig. 5i–k). In contrast to WT FL HP1α, NMR resonances of CSD in the $^{1}$H,$^{15}$N HSQC spectrum of FL L139E/L150E HP1α became detectable, indicating that mutation of the β-sheet interface residues L139 and L150 disrupted the formation of higher order HP1α oligomers (Fig. 5i). The concentration-dependent oligomerization observed for WT FL HP1α in MP assays (Fig. 5c, top two panels) was not observed for FL L139E/L150E HP1α (Fig. 5c, bottom two panels), corroborating NMR results that the hydrophobic β-sheet interface residues are necessary for HP1α oligomerization. We note that mutations of the β-sheet interface residues, such as L139, were introduced in previous studies with the goal to reduce oligomerization of HP1α$_{CSD}$[26,37].

Binding of the KAP1$_{Hbox}$ peptide further increased oligomerization of WT FL HP1α in MP assays, however the KAP1$_{Hbox}$-promoted oligomerization was markedly decreased in the case of the L139E/L150E mutant of FL HP1α (Fig. 5j). The inability of KAP1$_{Hbox}$ to augment oligomerization of L139E/L150E HP1α was not due to the impaired binding of the peptide. Titration of KAP1$_{Hbox}$ into $^{15}$N-labeled L139E/150E HP1α$_{CSD}$ led to CSPs that as for WT HP1α$_{CSD}$ were in the slow exchange regime, indicating that the replacement of L139 and L150 does not disrupt the CSD-KAP1$_{Hbox}$ complex formation (Fig. 5k). Collectively, these data revealed that while HP1α$_{CSD}$ forms a dimer through α-helices, HP1α oligomerization involves the β-sheet of HP1α$_{CSD}$. Interestingly, previous studies have demonstrated that mutations of the residues in either α-helix interface or β-sheet interface of HP1α$_{CSD}$ attenuate its repressive function in a luciferase reporter assay[26].

## Binding of KAP1$_{Hbox'}$ leads to a symmetric arrangement of HP1α$_{CSD}$

Overlay of the structures of HP1α$_{CSD}$ in the apo-state (yellow) and HP1α$_{CSD}$ (pink) bound to KAP1$_{Hbox'}$ within the larger crystal lattice showed that while the protomers HP1α$_{CSD1}$ and HP1α$_{CSD2}$ of a dimer can be superimposed, the protomers from the neighboring dimers were oriented distinctly differently in the crystal structure (Fig. 6a and Supplementary Fig. 9b, c). The orientation of one of the neighboring dimers in the apo-state structure was opposite to its orientation in the complex and overall, the neighboring dimers in the apo state structure were not superimposable (Fig. 6a–c, only one protomer from the neighboring dimers is shown in (a) and (b) for clarity). In contrast, binding of KAP1$_{Hbox'}$ led to a rigid complex characterized by a twofold symmetry and identical β-interfaces (Fig. 6b, c). MD simulations of the HP1α$_{CSD}$ protomers with KAP1$_{Hbox'}$ or removing KAP1$_{Hbox'}$ corroborated structural analysis, revealing significant distance deviations at the interfaces in the KAP1$_{Hbox'}$(-) state (Fig. 6d). Interestingly, incorporating additional symmetry mates from the crystal lattice showed that the HP1α$_{CSD}$-KAP1$_{Hbox'}$ complex crystallizes in an oligomeric spiral form with a diameter of ~66 Å and the vertical step of ~67 Å, large enough to potentially wrap around two double-stranded strands of DNA (Fig. 6e and Supplementary Fig. 11). The positively charged side chain of R487 is oriented toward the inner side of the spiral and therefore may augment binding to DNA. Oligomerization of HP1α was further increased by FL KAP1, as MP measurements collected on the mixture of FL HP1α and FL KAP1 showed mass distribution of multimeric species (Fig. 6f–i and Supplementary Fig. 12). In addition to the expected complexes formed between dimeric KAP1[38] and either two or four molecules of HP1α or between dimeric HP1α and one molecule of KAP1, we also observed higher order multimers (>300 kDa) (Fig. 6f, g and Supplementary Fig. 12). We speculate that this model can potentially promote the closed state of chromatin and reduce its accessibility, which is observed experimentally at ERVs.

## The direct HP1-KAP1 interaction contributes to the closed chromatin conformation at ERVs

Reduced accessibility is a feature of heterochromatin that contributes to silencing of specific genomic regions, and KAP1/SETDB1 complexes are known to maintain their targets in an inaccessible chromatin conformation[39,40], and we confirmed that HP1 is also required to maintain closed chromatin at retroelements in ESCs (Supplementary Fig. 13). To assess whether the direct interaction of KAP1 with HP1 contributes to the KAP1- and HP1-dependent inaccessibility at these regions, we developed a model in ESCs in which WT KAP1 or its V488E mutant, impaired in binding to HP1, were expressed under the control of a doxycycline-inducible promoter in the presence or absence of endogenous KAP1 (Fig. 7a). We first validated this model through demonstrating that the deletion of KAP1 results in loss of H3K9me3 heterochromatin and transcriptional activation as indicated by gained

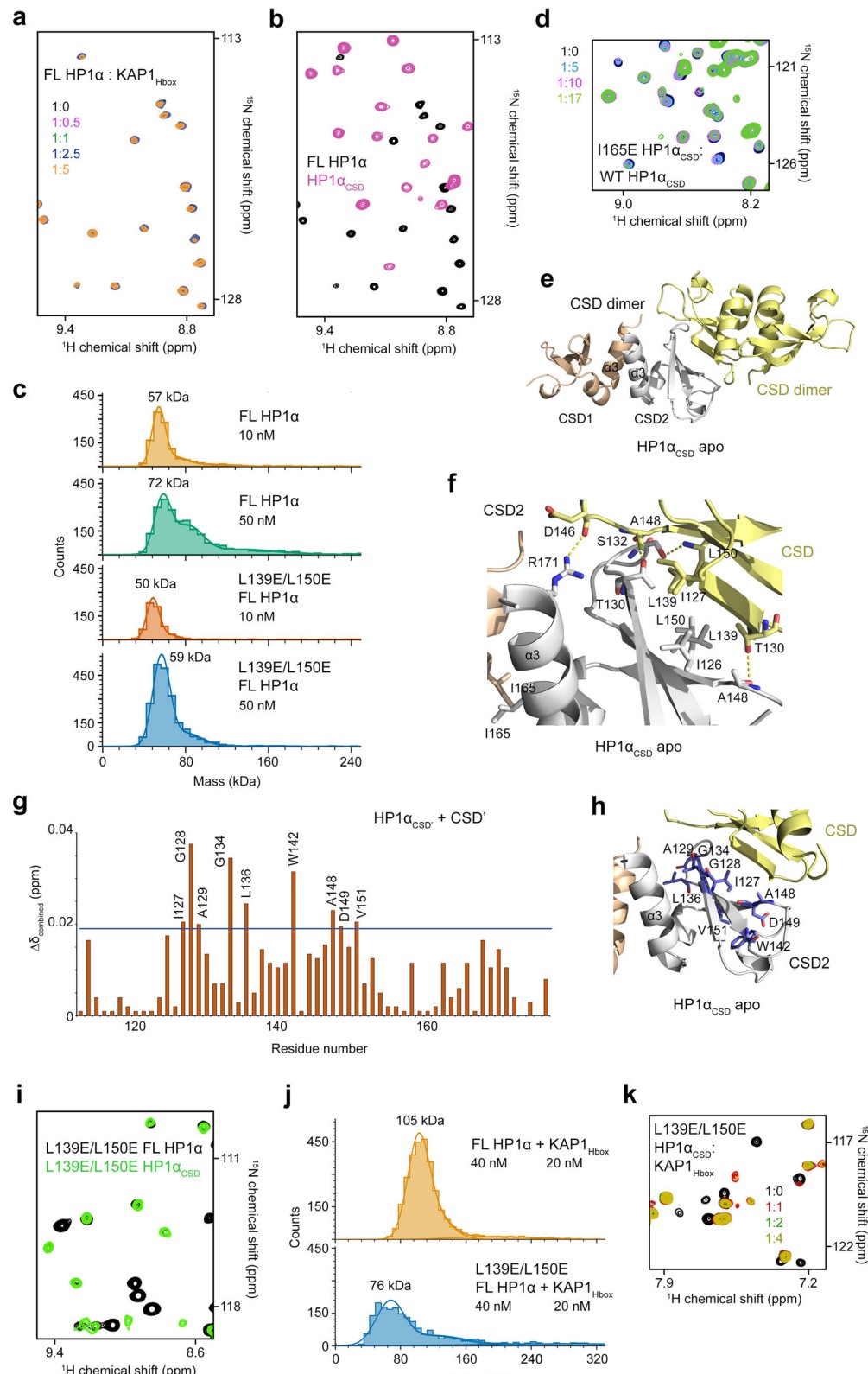

H3K27ac enrichment at a subset of endogenous retroelements (e.g., IAPEz, MERVK10C, and ETn) in ESCs (Fig. 7b–e). We next performed ATAC-seq on wild type (WT) and KAP1 conditional knock out (cKO) ESCs, as well as on KAP1 cKO ESCs expressing either exogenous WT KAP1 or the V488E mutant of KAP1. We found that the elements that were most enriched with KAP1, including IAPEz, MERVK10C and ETn (Fig. 1g and Supplementary Fig. 1), show strong reliance on its presence

to maintain an inaccessible chromatin state (Fig. 7b, f–h). Exogenous expression of KAP1 in KAP1 cKO ESCs was able to partially restore closed chromatin at IAPEz, MERVK10C and ETn elements. Furthermore, we found that the HP1-KAP1 interaction contributes to closed chromatin at ERVs, as the expression of the V488E mutant of KAP1, incapable of binding to HP1, could not restore a closed chromatin state to the same extent observed with expression of WT KAP1 in KAP1 cKO

**Fig. 5 | HP1α oligomerizes through HP1α$_{CSD}$. a** Overlayed $^1$H,$^{15}$N HSQC spectra of $^{15}$N-labeled FL HP1α collected in the presence of increasing amounts of KAP1$_{Hbox}$ peptide. Spectra are color coded according to the protein:peptide molar ratio. HP1α$_{CSD}$ signals in FL HP1α are broadened beyond detection, indicating that the size of this part of HP1α becomes too large, i.e. HP1α$_{CSD}$ multimerization (higher order than detectable dimerization) reduces tumbling rate. **b** Superimposed $^1$H,$^{15}$N HSQC spectra of $^{15}$N-labeled FL HP1α (black) and HP1α$_{CSD}$ (pink). **c** Molecular mass distribution histograms of FL WT and L139E/L150E HP1α at indicated concentrations in mass photometry assay. Maxima of the fits are labeled (kDa). **d** Overlayed $^1$H,$^{15}$N HSQC spectra of $^{15}$N-labeled I165E HP1α$_{CSD}$ collected in the presence of increasing amounts of unlabeled WT HP1α$_{CSD}$. Spectra are color coded according to the molar ratio of CSDs. **e** The crystal structure of the apo-state of HP1α$_{CSD}$ is depicted in a ribbon diagram (two dimers form an asymmetric unit, see Supplementary Fig. 9a). Two protomers of one HP1α$_{CSD}$ dimer (CSD1 and CSD2) are colored wheat and grey, and both protomers of the neighboring dimer are colored yellow. The dimers interact through their β-sheets. **f** A close view of the β-sheet binding interface, with the residues involved in the contact shown as sticks and labeled. Dashed lines represent hydrogen bonds. **g** Bar plot of resonance changes in $^1$H,$^{15}$N HSQC spectra of $^{15}$N-labeled HP1α$_{CSD'}$ induced by the 20-fold molar excess of unlabeled HP1α$_{CSD'}$. **h** Most perturbed in (**g**) residues are mapped on the structure of the apo-HP1α$_{CSD}$, colored blue and labeled. **i** Superimposed $^1$H,$^{15}$N HSQC spectra of $^{15}$N-labeled FL L139E/L150E HP1α (black) and L139E/L150E HP1α$_{CSD}$ (light green). **j** Molecular mass distribution histograms of FL WT and L139E/L150E HP1α in the presence of KAP1$_{Hbox}$ at indicated concentrations in mass photometry assays. Maxima of the fits are labeled (kDa). **k** Overlayed $^1$H,$^{15}$N HSQC spectra of $^{15}$N-labeled L139E/L150E HP1α$_{CSD}$ recorded in the presence of increasing amounts of KAP1$_{Hbox}$ peptide. Spectra are color coded according to the protein:peptide molar ratio.

ESCs (Fig. 7b–h). Reliance on the KAP1-HP1 interaction for maintaining inaccessible chromatin positively correlated with KAP1 enrichment at IAPEz, MERVK10C and ETn elements, positively correlated with increased H3K27ac in the absence of KAP1, and negatively correlated with loss of H3K9me3 in the absence of KAP1 (Fig. 7i–k). These observations did not hold true for retroelements that were not enriched with KAP1 (Supplementary Fig. 14). Collectively, these results indicated that in addition to KAP1's known role in HP1 recruitment through H3K9me3, the direct binding of KAP1 to HP1 has a role in silencing of ERVs in ESCs.

DNA sequences of ERVs and other retrotransposons, which comprise ~35% of the human genome, must be recognized by host defense mechanisms to prevent expression of these elements and ensure longstanding genomic stability[41]. Heterochromatization of the foreign DNA territories encoding retrotransposons occurs early in development and is maintained throughout the lifespan. Weakened heterochromatization of these DNA territories leads to transcriptional derepression of retrotransposons and is linked to the development of neurological and autoimmune diseases and cancer and is commonly associated with ageing[41]. KAP1, SETDB1, H3.3, H3K9me3 and HP1 proteins represent the core of the machinery to render DNA of ERVs in an inaccessible state, and it has been shown that repression of some retrotransposon families, while being independent of DNA methylation, depends highly on KAP1[16,42]. KAP1 itself forms a dimer though the central long coiled-coil region and interacts with KRAB domain containing zinc-finger proteins that facilitate recruitment to specific DNA sequences[38,43–47]. The C-terminal PHD finger-bromodomain cassette of KAP1 facilitates the recruitment of SETDB1 and the NURD/HDAC complex which all are required for the efficient repressive activity of KAP1[48]. Our study suggests a mechanism by which binding of KAP1 to HP1 can provide an extra level of control of transposons' DNA territories and stabilization of heterochromatin through altering electrostatic interactions and inducing conformational changes. The HP1 binding motif of KAP1, which contains an exposed DNA-recognizing arginine and is surrounded by the positively charged residues, appears to not only organize HP1α$_{CSD}$ but also improves the engagement of HP1α with DNA. This sequence of KAP1 differs from the sequences of other reported ligands of HP1, including CAF-1, Sgo1 and Sp100A that are mostly hydrophobic in nature and lacking the exposed arginine. Selective binding of KAP1 may promote the high degree of co-occupancy of HP1 and KAP1 at specific genomic regions that are needed to be tightly condensed, such as ERVs. In support, our in vivo model derived from ESCs shows that the KAP1-HP1 complex formation is required for chromatin silencing at ERVs.

## Methods
### Cell culture
ESCs were maintained under standard conditions on gelatin-coated plates at 37 °C at 5% CO2 in Knockout DMEM (Thermo Fisher) supplemented with NEAA, GlutaMAX, penicillin/streptomycin (Thermo Fisher), 15% ESC-screened fetal bovine serum (Hyclone), 0.1 mM 2-mercaptoethanol (Fisher), and leukemia-inhibitory factor (LIF). Generation of KAP1 cKO ESCs has been described previously[16]. The piggybac transposon system was used to stably integrate transgenic Flag-tagged WT or V488E mutant KAP1 under control of an rtTA promoter (Systems Biosciences, PB-TRE-EGFP-EF1a-rtTA) into the inducible KAP1 cKO ESCs. Exogenous expression of WT or V488E mutant KAP1 was induced with 2 μg/mL doxycycline for 48 h and maintained in doxycycline prior to endogenous KAP1 deletion. KAP1 deletion was induced with 2 μM 4-OHT (Sigma) treatment for 48 h. ESCs were routinely screened for mycoplasma. HCT116 parental cells were cultured in Dulbecco's modified Eagle's medium (DMEM) (Gibco, catalog 11965118) supplemented with 7% Fetal Bovine Serum (FBS) (MilliporeSigma, catalog F4135) and 1% Penicillin/Streptomycin (P/S) (Gibco, Cat# 15140163) at 37 °C with 5% CO2. The original HCT116 cell line was purchased from ATCC (Cat# CCL-247). Mycoplasma tests (SouthernBiotech, catalog 13100-01) were conducted every ~3–6 months.

### ChIP-seq
Native ChIP was performed with 5×106 cells. Spike-In chromatin (Active Motif, 53083) and Spike-in antibody (Active Motif, 61686) were used according to manufacturer instructions. Cells were trypsinized, washed and subjected to hypotonic lysis (50 mM TrisHCl pH 7.4, 1 mM CaCl2, 0.2% Triton X-100, 10 mM NaButyrate, and protease inhibitor cocktail (Roche)) with micrococcal nuclease for 5 min at 37 °C to recover mono- to tri-nucleosomes. Nuclei were lysed by brief sonication and dialyzed into RIPA buffer (10 mM Tris pH 7.6, 1 mM EDTA, 0.1% SDS, 0.1% Na-Deoxycholate, 1% Triton X-100) for 2 h at 4 °C. Soluble material was incubated with 3–5 μg of antibody (H3K9ac: Abcam, Cat# 4441, H3K27ac: Active Motif, Cat# 39133, H3K27me3: Cell Signaling, Cat# 9733) bound to 50 μl protein A or protein G Dynabeads (Invitrogen) and incubated overnight at 4 °C, with 5% reserved as input DNA. Magnetic beads were washed as follows: 3× RIPA buffer, 2× RIPA buffer +300 mM NaCl, 2× LiCl buffer (250 mM LiCl, 0.5% NP-40, 0.5% NaDeoxycholate), 1× TE + 50 mM NaCl. Chromatin was eluted and treated with RNaseA and Proteinase K. ChIP DNA was purified and dissolved in H2O.

ChIP-seq libraries were prepared from 5 to 10 ng ChIP DNA following the Illumina TruSeq protocol. The quality of the libraries was assessed using a D1000 ScreenTape on a 2200 TapeStation (Agilent) and quantified using a Qubit dsDNA HS Assay Kit (Thermo Fisher). Libraries with unique adaptor barcodes were multiplexed and sequenced on an Illumina NextSeq 500 (paired-end, 33 base pair reads). Typical sequencing depth was at least 20 million reads per sample.

### ATAC-seq
The modified ATAC-sequencing protocol Omni-ATAC was performed. Briefly, nuclei from 10$^6$ cells were made using NE1 buffer (20 mM HEPES pH 8, 10% Glycerol, 0.34 M Sucrose, 10 mM KCl, 2 mM EDTA,

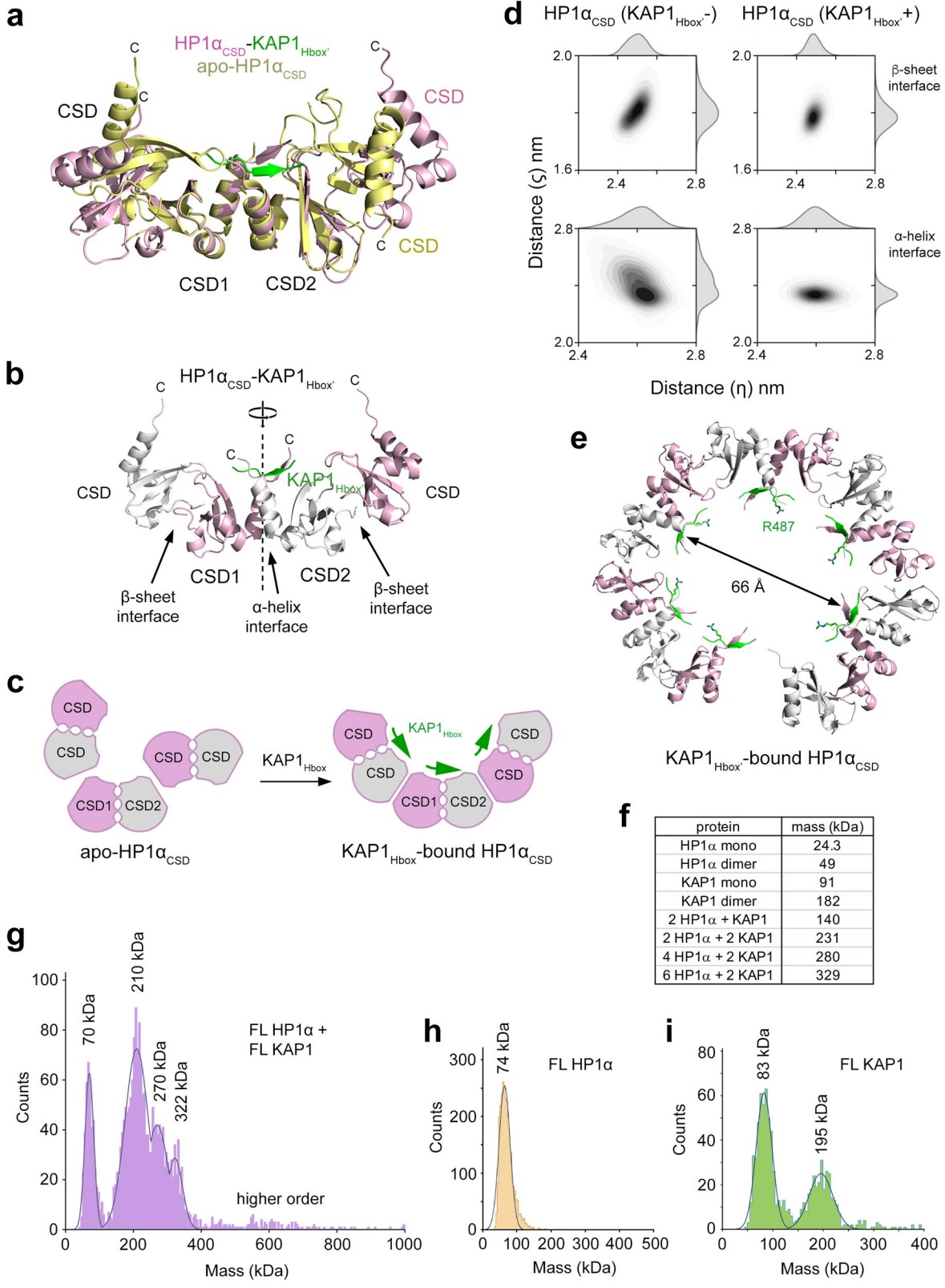

10 mM Sodium butyrate, 1× Roche Complete Protease Inhibitor, 0.5 mM Spermidine). Nuclei were washed once in NE1 then 6.5 ×10⁴ nuclei per replicate were resuspended in tagmentation buffer (25 µL 2× Tagment DNA Buffer (Illumina Cat# 20034197), 16.5 µL PBS (filtered), 4.875 µL water, 0.625 µL 8% Tween-20, 0.5 µL 1% Digitonin, 2.5 µL TDE1 Tagment DNA Enzyme (Illumina Cat# 20034197), 100 nM Tn5 final) at 37 °C for 30 minutes. The reaction was immediately purified using

Qiaquick PCR Purification Kit (Qiagen) and eluted in 21 µl water. Eluted DNA was amplified for 5 cycles using NEXTERA i7/i5 primers and NEBNext Ultra II PCR Master Mix (NEB). Cycle 1: 72 °C 5 min (gap filling), Cycle 2: 98 °C 30 s, Cycle 3: 98 °C 10 s, Cycle 4: 63 °C 10 s, Repeat Cycles 3-4. Number of further PCR cycles required was determined using KAPA Library Quantification Kit (KK4854). Typically ATAC libraries required 7–9 total cycles. Libraries were purified with AMPure

**Fig. 6 | Binding of KAP1Hbox' leads to a symmetric arrangement of HP1αCSD.**
**a** Overlayed structures of HP1$_{CSD}$ (pink) in complex with KAP1$_{Hbox'}$ (green) and of apo-state of HP1$_{CSD}$ (yellow). CSD1 and CSD2 protomers but not the neighboring dimers can be superimposed. Only one protomer from each neighboring dimer (labeled as CSD) is shown for clarity. **b** A ribbon diagram of the HP1$_{CSD}$-KAP1$_{Hbox'}$ complex. The protomers of a dimer (CSD1 and CSD2), the protomers of the neighboring dimers (CSD), the α-helix interface, and the β-sheet interfaces are labeled. Only one protomer from each neighboring dimer is shown for clarity. **c** A model of the structural reorganization of HP1$_{CSD}$ upon binding of KAP1$_{Hbox'}$. **d** MD simulation analysis of structural stability of HP1$_{CSD}$. Kernel Density Estimation (KDE) plots showing the distribution of distances η and ζ between the center of

mass of the first monomeric unit and the Cα atoms of residues 123 and 136 at the α-helix and β-sheet interfaces in HP1$_{CSD}$ without and with KAP1$_{Hbox'}$. **e** Structural organization of the multimer of the HP1$_{CSD}$-KAP1$_{Hbox'}$ complex. The positively charged side chain of R487 of KAP1$_{Hbox'}$, shown as a green stick, is oriented toward the inner side of the spiral. **f** Theoretical molecular masses of indicated proteins and complexes. **g** Molecular mass distribution histogram of the 1:1 mixture of FL HP1α and FL KAP1 in mass photometry assay. Maxima of the fits are labeled (kDa). Molecular mass distribution histograms of FL HP1α (**h**) and FL KAP1 (**i**) in mass photometry assay. Maxima of the fit are labeled (kDa). Source data are provided as a Source Data file.

XP beads (Beckman Coulter, Cat# A63881) using a double-sided selection strategy (0.6× followed by 1.3×). Samples were pooled for multiplexing and sequenced using 50 bp paired-end sequencing on Illumina NextSeq 500.

## CUT&Tag
For CUT&Tag nuclei were prepared as described above (ATAC-seq). $2.5 \times 10^5$ nuclei per replicate was resuspended in Wash125 buffer (20 mM HEPES pH 7.5, 125 mM NaCl, 10 mM Sodium butyrate, 0.025% Digitonin, 1x Roche Complete Protease Inhibitor, 0.5 mM Spermidine) and bound to CUTANA Concanavalin A Beads (Epicypher, Cat# 21-1401) for 15 min at 4 °C. Bound nuclei were then incubated with 50 μL Wash125 + 0.1% BSA, 2 mM EDTA, and 1 μL Primary antibody (H3K9me3: Abcam, Cat# 8898, H3K27ac: Active Motif, Cat# 39133) overnight at 4 °C. Nuclei were washed once with Wash125 and resuspended in 100 μL Wash125 + 1 μL Secondary antibody at room temperature for 60 min at 4 °C. Nuclei were washed twice in 1 mL Wash125 then resuspended in 200 μL Wash125 + 0.2% formaldehyde for 2 min at RT. After 2 min, formaldehyde was quenched with 50 μL of 2.5 M Glycine. Nuclei were washed once in 1 mL Wash350 (20 mM HEPES pH 7.5, 350 mM NaCl, 10 mM Sodium butyrate, 0.025% Digitonin, 1× Roche Complete Protease Inhibitor, 0.5 mM Spermidine) then incubated in 47.5 μL Wash350 and 2.5 μL pAG-Tn5 (Epicypher, Cat# 15-1017) for 60 min at RT. Nuclei were washed twice in 1 mL Wash350, then resuspended in 300 μL Wash350 and 10 mM MgCl2 and incubated for either 120 min (H3K9me3) or 60 min (H3K27ac) at 37 °C. Tn5 reaction was stopped with 10 μL 0.5 M EDTA, 3 μL 10% SDS, and 3 μL 18 mg/mL Proteinase K, briefly vortexed, then incubated at 55 °C for 120 min to reverse crosslinks and release fragments. The fragments were then purified with phenol-chloroform and resuspended in 22 μL 1 mM Tris-HCl pH 8, 0.1 mM EDTA. The entire sample was amplified with NEXTERA i7/i5 primers and NEBNext High-Fidelity Master Mix as described above. Typically CUT&Tag libraries required 8–12 total cycles. Libraries were purified with AMPure XP beads (Beckman Coulter, Cat# A63881) using a double-sided selection strategy (0.6× followed by 1.3×). Samples were pooled for multiplexing and sequenced using 150 bp paired-end sequencing on Novaseq 6000.

## Data processing
Quality of datasets was assessed using the FastQC tool. Raw reads were adapter and quality trimmed using Trimgalore. Trimmed reads were aligned to the mouse reference genome (mm10) with Bowtie2. Multimapping reads were randomly assigned. Optical duplicate reads were filtered using Picard (https://broadinstitute.github.io/picard/). Reads which mapped to the mitochondrial genome were removed with Samtools. Peak calling was performed with MACS2 software (broad peakcalling was used for H3K9me3). Peaks which intersected blacklisted high-signal genomic regions were removed. For merging of replicates, each replicate was downsampled to the depth of the lowest replicate then merged using Picard. BigWig files were generated from alignments using deepTools and normalized to counts per million (CPM). Visualization of merged bigWigs was done in Integrative

Genomics Viewer. Intersections between different peak sets were made using BEDTools. Heatmaps and average profiles were generated using deepTools. For correlation plots, density maps were then generated using ggplot2. Rho and p-values were obtained using cor.test(x, y, method = "spearman", exact = TRUE).

## Repetitive element analysis
Annotations for the genomic locations of repetitive element subfamilies were obtained from Dfam's non-redundant hits files (mm10.nrph.hits.gz)[49]. From these, separate bedfiles were generated for each subfamily. For heatmaps (Fig. 1g and Supplementary Fig. 1), counts were generated for each dataset at each subfamily using featureCounts[50]. Low-coverage loci with less than 1 input read were removed. Depth-normalized ChIP reads over depth-normalized input reads were calculated at each annotated locus of each repetitive element. Enrichment for each subfamily is shown as the median ChIP signal over input. To allow better comparison across data sets, the median enrichment values of all 1380 mm10 repetitive element families were z-score transformed per ChIP (i.e., each data set was normalized to itself). These z-scores were then plotted using the R package pheatmap (https://github.com/raivokolde/pheatmap). For coverage heatmaps and average profiles, bedfiles were generated from the LTRs of the indicated families such that if both the 5' and 3' LTR were present at the same genomic integration then only the 5' LTR would be represented (IAPEz: IAPLTR1_Mm and IAPLTR2_Mm; MERVK10C: RLTR10C, ETn: RLTRETN_Mm; MERVL: MT2_Mm). Plots of these families were then generated −1500 bp and +8000 bp from these LTR beds using deepTools computeMatrix and plotHeatmap. Average profiles were generated from ATAC datasets at these same LTR bedfiles. p-values were generated by counting reads within these LTRs using R countOverlaps then performing pairwise Wilcoxon rank-sum tests which were further adjusted by Benjamini & Hochberg procedure.

## Protein production and purification
Human full-length HP1α and HP1$_{CSD}$ (aa 109–185 of HP1α) were cloned into a pGEX4T-1 expression vector and expressed in *E. coli* BL21 Rosetta (DE3) pLysS cells grown in Luria Broth or M9 minimal media supplemented with $^{15}$NH$_4$Cl (Sigma-Aldrich). Following induction with 0.5 mM isopropyl β-D-1-thiogalactopyranoside (IPTG) for 20 h at 16 °C, cells were harvested by centrifugation and lysed by sonication. GST-tag fusion proteins were purified on glutathione agarose 4B beads (Thermo Fisher Sci) in 50 mM Tris-HCl (pH 7.5) buffer, supplemented with 500 mM NaCl and 5 mM β-mercaptoethanol. The GST tag was cleaved with TEV (Tobacco Etch Virus) protease overnight at 4 °C. Proteins were further purified by size exclusion chromatography (SEC) and concentrated in Millipore concentrators. Mutants were generated by site-directed mutagenesis using the Stratagene QuikChange mutagenesis protocol and were purified as wild-type proteins.

For EMSA, KAP1$_{Hbox'}$ (aa 468–496) was cloned in pCIOX expression vector with N-terminal His-SUMO tag and Ulp1 cleavage site. The

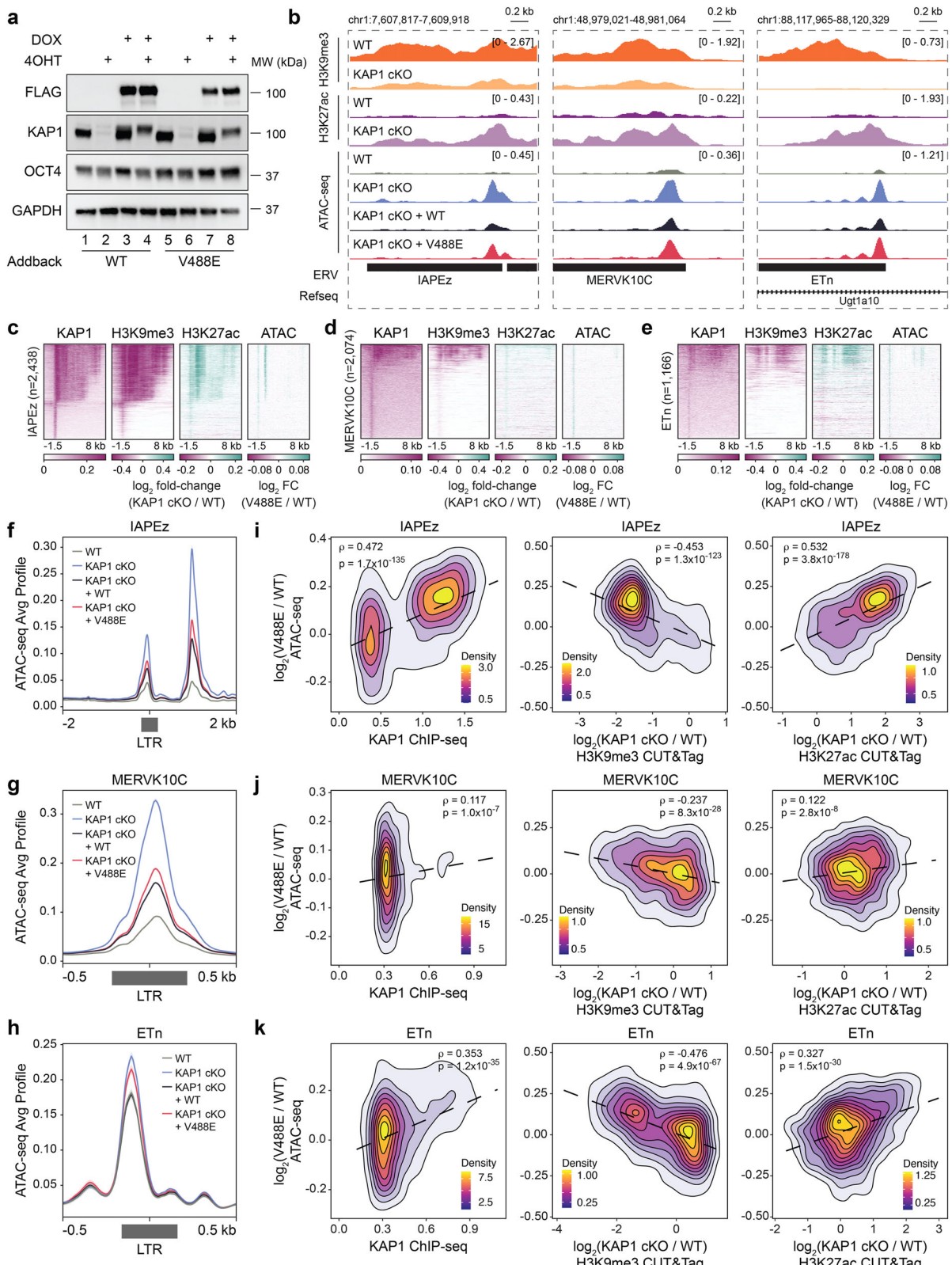

protein was expressed in *E. coli* BL21(DE3) RIL competent cells in Luria Broth, induced with 0.5 mM IPTG and grown for 20 h at 16 °C. The cells were harvested by centrifugation and lysed by sonication. His-SUMO tagged $KAP1_{Hbox}$ was purified on Ni-NTA beads (Qiagen) in 50 mM Tris-HCl (pH 7.5) buffer, 500 mM NaCl and 5 mM Dithiothreitol (DTT). The protein was eluted in 50 mM Tris-HCl (pH 7.5) buffer supplemented with 500 mM NaCl, 500 mM imidazole and 5 mM DTT. The protein was

further purified by size exclusion chromatography using a HiLoad16/600 Superdex 75 column (Cytiva) and buffer containing 50 mM Tris-HCl (pH 7.5), 500 mM NaCl and 5 mM DTT. The eluted protein fractions were analyzed by SDS-PAGE. The fractions were pooled, concentrated and exchanged with buffer containing 25 Tris-HCl pH 7.5, 150 mM NaCl and 2.5 mM DTT. The His-SUMO tag was cleaved overnight at 4 °C with Ulp1 enzyme and the tag was removed using Ni-NTA

**Fig. 7 | The direct HP1-KAP1 interaction contributes to inaccessible chromatin at ERVs in ESCs. a** Immunoblot of whole cell lysates from *Kap1fl/fl;Cre-ERT2* ESCs expressing either exogenous KAP1 or a KAP1 V488E mutant under control of a tetracycline response element. ESCs were treated with either DMSO or 2 µg/mL doxycycline for 48 h followed by treatment with EtOH or 2 µM 4-Hydroxytamoxifen for 48 h. *n* = 3 with identical results. **b** Genome browser representations of H3K9me3 and H3K27ac CUT&Tag in WT and KAP1 cKO ESCs as well as ATAC-seq in WT and KAP1 cKO ESCs and KAP1 cKO ESCs expressing exogenous KAP1 or a KAP1 V488E mutant at representative ERVs. The y axis represents read density in counts per million (CPM) mapped reads for an experiment performed in triplicate. **c–e** Heatmaps of KAP1 enrichment, the relative H3K9me3 and H3K27ac CUT&Tag enrichment difference between WT and KAP1 cKO, and the relative ATAC-seq enrichment difference between KAP1 cKO ESCs expressing either WT or KAP1

V488E at **c** IAPEz (*n* = 2438), **d** MERVK10C (*n* = 2074), and **e** ETn (*n* = 1166) ERVs. **f–h** ATAC-seq average profiles in WT and KAP1 cKO ESCs and KAP1 cKO ESCs expressing exogenous KAP1 or a KAP1 V488E mutant at LTRs of **f** IAPEz *n* = 2438, *p* = 0.0004), **g** MERVK10C (*n* = 2074, *p* = 7.94 × 10⁻¹²) and **h** ETn (*n* = 1166, *p* = 0.0003)). LTRs were either orphan or adjacent to full-length elements as defined in the methods. Significance determined by Wilcoxon rank sum with BH correction. **i–k** Correlation plots between differential ATAC-seq signal in KAP1 cKO ESCs expressing either WT or KAP1 V488E compared to KAP1 ChIP-seq (left), differential H3K9me3 CUT&Tag in KAP1 cKO compared to WT ESCs (middle), and differential H3K27ac CUT&Tag in KAP1 cKO compared to WT ESCs (left) at **i** IAPEz (*n* = 2438), **j** MERVK10C (*n* = 2074), and **k** ETn (*n* = 1166). Spearman correlation tests were used to determine statistical significance. Source data are provided as a Source Data file.

beads. The purified KAP1H$_{box'}$ was concentrated in Millipore concentrators.

For droplet assays, cDNA of full-length HP1α was cloned into a pET28 expression vector. His-tag fusion HP1α was expressed in *E. coli* and purified as in ref. 51. cDNA of the KAP1$_{FR}$ fragment (aa 114–834) was cloned into a pCAG-GFP expression vector, and the P486A/V488A/L490A mutant of GFP-KAP1$_{FR}$ was generated using overlap extension PCR. Purification of GFP-KAP1$_{FR}$ proteins was described in ref. 52.

### Fluorescence spectroscopy

Spectra were recorded at 25 °C on a Fluoromax Plus-C spectrofluorometer (HORIBA) as described[53] with the following modifications. The samples containing 1 µM of HP1α$_{CSD}$ in 20 mM Tris pH 7.4, 200 mM NaCl, 2 mM DTT and progressively increasing concentrations of KAP1$_{Hbox}$ peptide (aa 483–493 of KAP1, synthesized by SynPeptide) were excited at 295 nm. Emission spectra were recorded between 320 and 360 nm with a 0.5 nm step size and a 0.6 s integration time and averaged over three scans. The K$_d$ value was determined using a nonlinear least-squares analysis and the equation:

$$\Delta I = \Delta I_{max} \frac{\left( ([L]+[P]+K_d) - \sqrt{([L]+[P]+K_d)^2 - 4[P][L])} \right)}{2[P]} \quad (1)$$

where [L] is the concentration of the peptide, [P] is the concentration of the protein, ΔI is the observed change of signal intensity, and ΔI$_{max}$ is the difference in signal intensity of the free and bound states of the protein. K$_d$ is the average of three separate experiments with error reported as SD.

### NMR experiments

NMR titration experiments were carried out at 298 K on Varian 600 MHz and 900 MHz spectrometers equipped with cryogenic probes. ¹H,¹⁵N HSQC spectra of 0.1–0.2 mM ¹⁵N-labeled wild type and mutant HP1α proteins (in 20 mM Tris pH 6.8, 150 mM NaCl, 2 mM DTT and 10% D₂O) were collected in the presence of increasing concentrations of KAP1$_{Hbox}$ peptide (aa 483–493 of KAP1, synthesized by SynPeptide) or HP1α$_{CSD}$. NMR data were processed and analyzed with NMRPipe[54] and NMRDraw as described[55]. Normalized chemical shift perturbations shown in Fig. 3b were calculated using the equation:

$$\Delta\delta = \sqrt{(\Delta\delta H)^2 + \left(\frac{\Delta\delta N}{5}\right)^2} \quad (2)$$

where Δδ is the change in chemical shift in parts per million (ppm).

NMR experiments for amide chemical shift assignments were collected at 310 K on an Avance Neo Bruker 800 MHz spectrometer equipped with a TXO cryo-probe optimized for ¹⁵N detection. The HP1α$_{CSD'}$ construct (aa 112–176 of HP1α) and the CD of HP1α (aa 18–75 of HP1α, HP1α$_{CD}$) were purified as described[15]. Experiments were

performed in buffer containing 20 mM HEPES (90% H₂O/10% D₂O), pH 7.2, 75 mM KCl, 1 mM TCEP, and 0.01% NaN₃. Chemical shift assignments were performed on a 400 µM uniformly ¹³C/¹⁵N-labeled HP1α$_{CSD'}$ sample using standard 2D ¹H,¹⁵N HSQC, 3D HNCACB and 3D CBCA(CO)NH experiments[56–58]. Data were processed with NMRPipe[54] and analyzed in NMRFAM-SPARKY[59] (Sparky). Backbone chemical shifts (amide proton (¹H), amide nitrogen (¹⁵N), alpha carbon (Cα), and beta carbon (Cβ)) were assigned for all residues except the N-terminal serine (present due to TEV cleavage), Pro 122, Pro 161, and Asp 112 (Cα and Cβ chemical shift assigned for Asp 112, Pro 122, and Pro 161).

For the chemical shift perturbation experiments shown in Fig. 5h and Suppl. Fig. 10, ¹H,¹⁵N HSQC spectra were recorded on 50 µM ¹⁵N-labeled HP1α$_{CSD'}$ samples with and without 1 mM unlabeled HP1α$_{CSD'}$ or with and without 900 µM unlabeled HP1α$_{CD}$. The resonance perturbation analysis was performed using the Hamming distance[60,61] as follows:

$$\Delta\delta_{combined} = |\Delta\delta_H| + \alpha|\Delta\delta_N| \quad (3)$$

The Δδ$_{combined}$ is the sum of the absolute chemical shift change. Δδ$_H$ is the change in the ¹H chemical shift. Δδ$_N$ is the change in the ¹⁵N chemical shift. α (0.105) is the scaling factor calculated by using the ratio between the backbone ¹H chemical shift range and ¹⁵N chemical shift range from the ¹H,¹⁵N HSQC spectrum.

### X-ray crystallography

HP1α$_{CSD}$ (aa 109–185) and KAP1$_{Hbox'}$ (aa 468–496) were co-expressed, and the complex was purified as above and concentrated to 20 mg/ml. Crystals of the HP1α$_{CSD}$-KAP1$_{Hbox'}$ complex were grown at 291 K using sitting drop vapor diffusion method. 1 µL protein solution was mixed with 1 µL reservoir containing 0.1 M Bis-Tris pH 5.8, 0.2 M MgCl₂, 20% PEG3350 and 30 mM glycyl-glycyl-glycine. Crystals were directly flash-frozen in liquid nitrogen. The apo-state of HP1α$_{CSD}$ was concentrated to 25 mg/ml. Crystals of the apo-state were obtained in the crystal screen containing 0.1 M HEPES pH 7, 0.2 M sodium chloride and 20% w/v PEG 6000. Although peptide was added during crystallization, no electron density was observed for the peptide. Crystals were flash-frozen in liquid nitrogen before collecting data. X-ray diffraction data were collected at 100 K on a Rigaku Micromax 007 high-frequency microfocus X-ray generator at the CU Anschutz X-ray crystallography core facility. XDS was used for indexing and integrating the datasets[62]. Scaling and merging were performed using Aimless, and the structures were solved by Phaser[63] in ccp4 using PDB ID: 3I3C as the search model. Model building was performed with Coot[64], and the structures were refined with Refmac5. The data processing and refinement statistics are summarized in Supplementary Table 1.

### Native-PAGE gel

Native-PAGE gel electrophoresis of full-length wild type and mutated HP1α was performed using 10 µL of 50 µM protein samples in 20 mM

Tris-HCl pH 7.5 buffer supplemented with 100 mM NaCl. Electrophoresis was performed at 80 V for 2 h in 0.5× Tris-borate-EDTA (TBE) buffer on ice. The gels were stained using Coomassie brilliant blue R-250.

## Strep affinity purifications (AP)

For all AP experiments, HCT116 cells were seeded in 10 cm plates (~4 million cells per dish) and transfected with Strep- and Flag-tagged plasmids one-day after seeding cells. Two days post-transfection, cells were collected in PBS and lysed with passive lysis buffer (PLB: 50 mM Tris-HCl pH 7.5, 150 mM NaCl, 1 mM DTT, 1.0% NP-40, 1.5 mM MgCl2, 5% v/v glycerol, and 1X complete, Mini EDTA-free, EASYpack Protease Inhibitor [Roche, Cat# 4693159001]) by resuspending cells in PLB, vortexing briefly, nutating for 15 min at 4 °C, and centrifuging the lysate (10,000 × $g$, 10 min, 4 °C). Strep-Tactin superflow resin (IBA Life Sciences, Cat# 2-1208-010), was equilibrated by centrifuging beads (2000 RPM, 2 min, 4 °C), removing storage buffer, and washing three times with PLB with nutation for 5 min at 4 °C. Beads were volumed back to their original volume, and 40 μL beads were used per AP and bound for 2 h. Beads were then washed with AP Wash Buffer (50 mM Tris-HCl pH 7.5, 250 mM NaCl, 1 mM DTT, 0.2% NP-40, 1.5 mM MgCl2, and 5% v/v glycerol), and eluted with Strep-tag Elution Buffer containing D-Desthiobiotin (IBA Life Sciences, Cat# 2-1000-025).

## Western blot

All western blots were run on 10-12% SDS-PAGE gels and transferred on nitrocellulose membranes (Bio-Rad, Cat# 1620115) using the Bio-Rad Trans-Blot Turbo Transfer System, blocked for 1 h in 5% Milk (or BSA) with Tris-buffered saline-Tween-20 (TBST), probed with primary antibody in 5% Milk (or BSA) with TBST. Blots were exposed using either Clarity Western ECL (Bio-Rad, Cat# 1705060) or utilizing the starbright secondary channel. KAP1 and CBX5/HP1a mutants for Western blot assays were generated using QuikChange II XL Site-Directed Mutagenesis kit (Agilent, Cat# 200522) per manufacturer's instructions. PCR amplified DNAs were transformed into *E. coli* BME treated XL-10 Gold provided in the kit and positive clones validated by Sanger Sequencing. Antibodies used: anti-KAP1 monoclonal antibody (Abcam, Cat# ab22553) at 1:2000 dilution, anti-FLAG M2 monoclonal antibody (Sigma, Cat# F1804) at 1:10000 dilution, HRP-conjugated Strep-Tag II monoclonal antibody (Millipore, Cat# 71591) at 1:10000 dilution and Goat anti-mouse IgG-HRP secondary antibody (Santa Cruz Biotechnologies, Cat# sc-2005) at 1:10000 dilution.

## EMSA

EMSA experiments were performed by mixing 0.25 pmol/lane of $DNA_{147}$ with an increasing concentration of full-length HP1α or $HP1α_{CSD}$ in the presence (1:10) and absence of $KAP1_{Hbox}$ peptide (aa 483–493 of KAP1, synthesized by SynPeptide) or $KAP1_{Hbox'}$ (aa 468–496 of KAP1) as well as with $KAP1_{Hbox}$ or $KAP1_{Hbox'}$ but without HP1α proteins. The buffer containing 25 mM Tris-HCl pH 7.5, 150 mM NaCl and 2.5 mM DTT was used to make up the final reaction volume of 10 μL. The samples were incubated for 1 h on ice and then loaded on a 8% native PAGE gel and run for 2 h at 80 V in 0.5× Tris-borate-EDTA (TBE) on ice. The gels were stained with SYBR Gold (Thermo Fisher Sci) and visualized by Blue LED (UltraThin LED Illuminator-GelCompany).

## Immunofluorescence

The constructs coding for $GFP-KAP1_{RF}$ and the P486A/V488A/L490A mutant were transfected into 1 ×10[5] KAP1[−/−] mouse ESCs using Lipofectamine 3000 (Invitrogen). Forty-eight hours post-transfection, cells were trypsinized and plated at a colony density in selection medium containing 10 μg/ml blasticidin (Thermo Fisher Sci) for one week. The selection process was repeated once to establish stable GFP-KAP1 expressing cell lines.

For immunostaining, mouse ESCs were cultured on Geltrex-coated coverslips (Life Technologies) in DMEM supplemented with 16% fetal bovine serum, 0.1 mM β-mercaptoethanol (Invitrogen), 2 mM L-glutamine, 1× MEM non-essential amino acids, 100 U/ml penicillin, 100 μg/ml streptomycin (PAA), 2i (1 μM PD032591 and 3 μM CHIR99021 (Axon Medchem, Netherlands) and 1000 U/ml recombinant LIF (Millipore). Coverslips were rinsed twice with PBS (pH 7.4; 140 mM NaCl, 2.7 mM KCl, 6.5 mM $Na_2HPO_4$, 1.5 mM $KH_2PO_4$), followed by fixation with 3.7% formaldehyde (Sigma) for 10 min. Fixed cells were washed three times (10 min each) with PBST (PBS + 0.01% Tween-20), permeabilized in 0.5% Triton X-100 for 10 min, and washed twice (10 min each) with PBST. Primary and secondary antibodies were prepared in blocking solution (PBST + 3% BSA). Cells were incubated sequentially with anti-HP1α primary antibody (Abcam, ab77256) at 1:250 dilution and anti-rabbit Alexa 647 secondary antibody (Invitrogen, A-21245) at 1:500 dilution for 1 h each in a humidified dark chamber, followed by three 10 min washes with PBST. DNA was counterstained using DAPI (400 ng/ml in PBST) for 10 min, followed by three additional 10 min washes with PBST. Coverslips were mounted in Vectashield antifade medium (Vector Laboratories) and sealed with colorless nail polish. Images were acquired using a Nikon TiE microscope equipped with a Yokogawa CSU-W1 spinning-disk confocal unit (50 μm pinhole size), an Andor Borealis illumination unit, an Andor ALC600 laser beam combiner (405 nm/488 nm/561 nm/640 nm), an Andor IXON 888 Ultra EMCCD camera, an Andor FRAPPA photobleaching module, and a Nikon 100×/1.45 NA oil immersion objective. The microscope was operated using Nikon NIS Elements software (ver. 5.02.00).

## Heterochromatin FRAP

E14 ESC cells were transfected with 3 μg of pdsRed-HP1α and either pGFP or $pGFP-KAP1_{FR}$ using AMAXA nucleofector (Lonza), seeded in slide chambers containing high precision glass overnight at 37 °C and 5% $CO_2$. Samples were analyzed using a Leica SP5-II confocal point scanner microscope equipped with 488 nm Argon ion and 561 nm DPSS 50 mW lasers, a AOBS beams splitter and a HyD Hybrid detector. Cells were first scanned for GFP[+] using a HCX PL APO 100x Corr CS objective, and a section of 64 × 64 pixels with 0.8-0.6 μm/px size containing one heterochromatin compartment of these cells was selected. Subsequently, a point region of interest was used to bleach the half of the heterochromatin compartment for 100 ms at maximum intensity of the laser (488 and 561 nm). The recovery was documented for 15 s with a frame rate of approximately 100 ms/frame. MOCHA-FRAP analysis was performed as in ref. [36].

## In vitro droplet assays

For the droplet assay, Widom 601 DNA was amplified by PCR using a plasmid coding for 12× 601 DNA with primers as follows: 601 DNA_F: GCCGCCCTGGAGAATCCC and 601 DNA_R: TGC ACA GGA TGT ATA TAT CTG ACA C. HP1α was concentrated to ~10 μg/μl using Amicon concentrators. The buffer changes for both HP1α and $GFP-KAP1_{FR}$ solutions were carried out using Zeba™ Spin Desalting Columns with a buffer containing 20 mM HEPES at pH 7.2, 75 mM KCl, and 1 mM DTT. For visualization of His-tagged-HP1α within the droplets, 500 ng of protein was labeled with the 647 N fluorescent label using the Monolith NT™ Protein Labeling Kit RED-NHS from Nano Temper. To form HP1α droplets, 50 μM of HP1α was incubated with 0.18 μM of 601 DNA for 3 min and dropped into ibidi chamber (μ-Slide 18 Well). Subsequently, 50 ng of the labeled HP1α and 100 ng of $GFP-KAP1_{FR}$ (wild type or mutant) were added into HP1α/DNA droplets. The images of the droplets were collected using a Nikon TiE microscope equipped with a Yokogawa CSU-W1 spinning-disk confocal unit (50 μm pinhole size), an Andor Borealis illumination unit, Andor ALC600 laser beam combiner (405 nm/488 nm/561 nm/640 nm), Andor IXON 888 Ultra EMCCD camera, and a Nikon 100×/1.45 NA oil immersion objective. The size

and intensity of HP1α phase-separated droplets were analyzed using ImageJ.

## MD simulations

The KAP1$_{Hbox}$-bound and free HP1α$_{CSD}$ structures were taken as starting models. MD simulations were performed using ff14SB force field[65] in GROMACS (v2018.6). The conformational sampling was enhanced by generalized Replica Exchange with Solute Tempering (gREST) method[66]. In the gREST, the dihedral-angle potential energy term of the selected solute region was scaled from 300 K to 600 K. The conformations sampled in the replica corresponding to 300 K were analyzed. The detailed MD protocol is provided in the Source Data file.

## Mass photometry

Mass photometry (MP) assays shown in Fig. 5c, j, were performed on a Refeyn Two$^{MP}$ mass photometer (Refeyn Ltd, Oxford, UK) to determine oligomeric states of FL and L139E/L150E HP1α. Each sample was loaded into the sample wells of the silicon cassettes assembled onto Mass-Glass UC coverslips (Refeyn, Ltd.). Samples for measurements were prepared by rapid in-drop dilution and measurements were performed at room temperature in buffer (50 mM Tris pH 7.5, 150 mM NaCl, 5 mM DTT). Before each measurement, 10 μl of buffer was added to a well, and the focus was set and locked. 10 μl of different concentrations of FL WT or L139E/L150E HP1α or the mixture of HP1α proteins and KAP1$_{Hbox}$ (2: 1 molar ratio) was added to the drop. Samples were briefly mixed, and movies were captured immediately for 60 s (2800 frames) using AcquireMP software (version 2024 R1, Refeyn Ltd). β-amylase in the same buffer was used as calibration standard. Data were processed using DiscoverMP (version 2024 R1, Refeyn Ltd).

For MP assays shown in Fig. 6g–I and Supplementary Fig. 12, FL KAP1 and FL HP1α were expressed and purified as described[38,67]. MP measurements were performed on a Refeyn OneMP instrument (Refeyn Ltd). The calibration was done with a native marker protein standard mix (NativeMark Unstained Protein Standard, Thermo Scientific), which contains proteins ranging from 20 to 1200 kDa. Coverslips (24 × 50 mm, No. 1.5H, Marienfeld) were cleaned by sequential sonication in Milli-Q water, isopropanol and Milli-Q-water, followed by drying with nitrogen. For each acquisition 2 μL of protein solution was applied to 18 μL of buffer (10 mM HEPES pH 7.5, 150 mM NaCl) in a gasket (CultureWellTM Reusable Gasket, Grace Bio-Labs) on a coverslip. To capture the movie, 400 nM HP1α was diluted 10× and 200 nM KAP1 was diluted 10× for a final concentration of 40 nM and 20 nM, respectively. The final molar ratios of HP1α:KAP1 were 1:0, 1:1 and 2:1. Movies were recorded at 999 Hz with an exposure time of 0.95 ms by using the AcquireMP software (Refeyn Ltd). All mass photometry movies were processed and analysed in the DiscoverMP software (Refeyn Ltd).

## Statistics and reproducibility

Tryptophan fluorescence data are presented as average of three independent measurements ±SD. Western blot experiments were performed in three biological replicates and showed identical results. EMSA experiments in Fig. 4 and Supplementary Figs. 6 and 7 were performed at least twice.

## Data availability

Coordinates and structure data generated in this study have been deposited in the Protein Data Bank under the accession codes 9CDW and 9CEA. The genomics data generated in this study have been deposited in NCBI's Gene Expression Omnibus and are accessible through GEO Series accession number GSE268208, GSE268209 and GSE293913. Other datasets used for this study are available under GSE59189 and GSE97945. All other relevant data supporting the key findings of this study are available within the article and its Supplementary Information files. Source data are provided with this paper.

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

## Acknowledgements

This work was supported in part by grants from the NIH: AG067664, HL151334 and CA252707 to T.G.K., GM124958 and HD109239 to L.A.B., AI114362 to I.D., R35GM138382 to G.T.D., and F99CA264296 to U.H. and from Welch (I-2025) to L.A.B. L.A.B is a Virginia Murchison Linthicum Scholar in Medical Research (UTSW Endowed Scholars Program), American Cancer Society Scholar (134230-RSG-20-043-01-DMC). This work was also supported by the Deutsche Forschungsgemeinschaft (DFG, German Research Foundation) grants CA198/16-1 project number 425470807 and CA198/19-1 project number 522122731 to M.C.C and grants LE 721/18-1 project number 425470807 and SFB 1064 Project-ID 213249687 to H.L., and by the AMED (BINDS, Basis for Supporting Innovative Drug Discovery and Life Science Research) grant number JP24ama121024.

## Author contributions

N.G., R.O., U.H., W.Q., C.H., H.R., A.K., M.J.M., R.K.S., R.H., K.S., J.L., S.M., Y.Y. and A.C. performed experiments and together with M.D.P., B.F., H.K., M.C.C., G.T.D., H.L., I.D., L.A.B. and T.G.K. analyzed the data. L.A.B. and T.G.K. wrote the manuscript with input from all authors.

## Competing interests

The authors declare no competing interests.
