## [Transparent Peer Review file · Nature Communications]

The HP1 box of KAP1 organizes HP1 α for silencing of endogenous retroviral elements in embryonic stem cells

Corresponding Author: Professor Tatiana Kutateladze

Version 0:

Reviewer comments:

Reviewer #1

(Remarks to the Author)

Gaurath et al. Report a structure-function study on the role of the HP1a interaction with KAP1 in silencing ERVs. While the functional studies appear to support such a role, the proposed mechanism, at least its structural aspect, is not clearly supported by the data.

Major concerns:

- The crystal structure of the apo form the CSD of HP1a shows a tetramer, and the NMR data is consistent with interactions other than the dimerization ones, as seen in the crystal. However, the affinity of this interaction is not quantified. Based on the magnitude of the CSP in Figure 5c this additional interaction seems to be very weak. The CSPs are about 10 times smaller than those caused by KAP1 peptide (even at lower molar ratios, figure 2c). As the interaction with the peptide is in the low microM range, the interaction with the -1 and +3 CSD protomers is, perhaps, in the millimolar range, questioning the significance of the tetramer seen in the crystal.

- Along the text, the symmetric nature of the complex between the CSD of HP1a and the peptide of KAP1 is stated and discussed. However, it seems to me that the complex is not symmetric. The dimeric structure of the CSD of HP1a in the apo form might be symmetric, but bound to the KAP1 peptide, as seen in the crystal structure, is asymmetric. The single KAP1 peptide as a beta strand parallel to one protomer and antiparallel to the other breaks the symmetry of the CSD dimer. NMR should be able to clarify this issue. The full HSQC spectrum of the CSD of HP1a in supp. Fig. 3 shows a single set of backbone amide signals, which is consistent with a symmetric oligomer, but the full spectrum of the CSD bound to the KAP1 peptide is not shown. If the complex is symmetric, a single set of signals should be observed. If it is asymmetric (as the crystal structure shows) two sets of signals should be observed, because of the magnetic inequivalence created by the single KAP1 peptide and the tight binding (in slow exchange). If only one set of signals is observed, it may be that the crystal structure does not adequately represent the complex in solution.

- Most of the panels in Figure 6, c are models built based on the interpretation of the crystal structures whose relevance must be reevaluated. Figure 6 is not a "diagram of the symmetrical tetramer of the HP1aCSD-KAP1Hbox complex" as said in the figure legend, or at least experimental evidence of the existence of this tetramer has not been presented.

- Overall, I think that the conclusion that binding to KAP1a leads to a symmetric rearrangement of HP1a and multimerization is not sufficiently supported by the data.

-The significance of some experiments is unclear. For instance, the assays on phase-separated samples shown in Figure 4c show that the concentration of HP1a in the droplets is higher when KAP1 is present than when absent (it recruits more HP1a to the droplets). This is consistent with KAP1 binding to DNA and HP1a. But what is the conclusion regarding the "phase separation ability of HP1a"? In the FRAP experiments, the dip is slightly higher when KAP1 is present, but it does not indicate that KAP1 enhances the ability of HP1a to create phase separation in ESCs.

- The experimental details in the methods section are insufficient to judge the experimental design and to reproduce the data. For instance, in mass photometry experiments, the volumes of samples are indicated, but not the relevant parameter, which is the protein concentration. This is just one example, not an exhaustive revision of the method details.

- The labeling of some figures is cryptic or unclear. For instance, "DIC" in Figure 4c, the masses expressed in percentage in Figures 6f and h, "normalized CSP" in several figures (in equation 2 there is no normalization, the factor α is not given and the calculated quantity is not named CSP).

I think that the authors should thoroughly re-evaluate the interpretation and presentation of the structural data as well as the description of the methods.

Reviewer #2

(Remarks to the Author)

I have read the manuscript titled "KAP1 organizes HP1 α for silencing of endogenous retroviral elements in embryonic stem cells". I am a researcher with more than a decade of experience working on TRIM28 and its role in epigenetic silencing of transposable elements. Their key result is the obtained structure of the KAP1 HP1 box peptide to a dimer of HP1, and some validation of established relationships between KAP1 and transposable elements. Here are a few comments that in my opinion would improve the text, as well as some concerns that I hope the authors will be able to address before further consideration for publication by the editor.

Major comments:

The authors need to tone down some claims, for example "Our findings provide mechanistic and functional insights essential in our understanding of how ERVs are silenced." could be "Our findings add some mechanistic and functional insights to our understanding of how ERVs are silenced." Another example: "we demonstrate the critical role of HP1 α and KAP1 in silencing of repetitive elements" would be more appropriate to say "we confirm". Please review every claim related to mechanisms of ERV silencing, for which there is a large body of literature and a common understanding of the mechanism implicating H3K9me3 recruiting HP1 to form liquid-like phase separated heterochromatin domains.

Another example: "The findings that KAP1-containing complexes mediate transcriptional repression of ERVs and KAP1 binds to HP1 proteins prompted us to investigate whether KAP1 and HP1 cooperate in ERVs silencing." Rephrase as "prompted us to dig deeper in the mechanisms to explain how KAP1 and HP1 cooperate to mediate ERV silencing".

Please rephrase the sentence at line 55 claiming "HP1 β and HP1 γ are implicated in both gene activation and repression, whereas

HP1 α is associated with transcriptional silencing and chromosome segregation and is enriched at pericentric heterochromatin". The 'activation' claim is poorly supported, many of the references cited are correlative (for example, with transcription elongation and not 'activation') and outdated, and many of which are in organisms where KAP1 doesn't exist.

Line 111: "To better understand the high degree of co-localization of KAP1 with HP1s at ERVs, we explored the underlying mechanism." Please discuss the commonly accepted mechanism of colocalization, which is simply that KAP1 leads to H3K9me3 deposition, and HP1 binds H3K9me3.

Line 118: Rephrase "and a larger region of KAP1" to "and a larger KAP1 fragment around the HP1 box"

In that section and all remaining ones, when referring to this larger KAP1 HP1 box peptide, please refrain from using 'KAP1' to refer to it - switch every instance of KAP1 (including in figures) to something more appropriately describing the peptide. A clearer naming scheme which leaves no ambiguity when experiments are done with the larger KAP1 hp1 box fragment is important. Referring to it as 'KAP1' is too ambiguous and potentially misleading.

Add a section discussing whether it is likely or not that the peptide would behave differently in the context of the full length protein (accessibility of various residues, surrounding charges, 3D conformation).

Similarly, add a section discussing whether the KAP1 HP1 box V488E mutant could have possible interactions / effects on other domains of the full-length protein?

"Unexpectedly, we found that while resonances of CD were observed in the 1H,15N HSQC spectrum of full-length HP1 α , most of the signals of HP1 α CSD were undetectable, even when the spectra were collected on a 900 MHz NMR spectrometer (Fig. 5a, b). These data suggested that HP1 α CSD may form oligomers larger than a dimer – HP1 α CSD is a small globular module that should be amenable to NMR studies in the dimeric form – and corroborated analytical ultracentrifugation studies describing HP1 α CSD dimer tetramer equilibrium." This section is arcane and non-descriptive to someone not familiar with NMR, please explain further in simpler terms.

Line 260 - ChIP-seq is notoriously non-quantitative, tone down conclusions that are weak based on quantification of partial complementation.

Line 268 - "Collectively, these results indicated that binding of KAP1 to HP1 has a role in silencing of ERVs in ESCs." This

is known for decades, use 'confirm' instead of 'indicated'.

While discussing their model of HP1 / KAP1 interaction in link with the observed tetramer structure, have the authors considered a simple model where a dimer of KAP1, each recruiting a dimer of HP1, explains the stoichiometry of HP1 oligomerization in presence of KAP1?

Figure 1g - please explain why there is no enrichment of any HP1 to sequences enriched in H3K9me3 and KAP1 - is the HP1 ChIP-seq less sensitive than others (signal-noise ratio)? Especially so that the signal is very much visible in the examples given in panel g? Is there some quantification mistake, for example counting all elements instead of just the bound ones?

Figure 4a - can this band shift be explained by high amounts of HP1 simply clogging the gel, especially so that it is known to form phase-separated domains at high concentrations? The bands are very smeary which would support this and not binding of HP1 to the DNA.

Figure 5c - is this 'tetramer' an artifact of crystallography (symmetry)? The outer 2 units mode of interaction is not particularly convincing - why are those outer units not dimerizing as well?

Minor comments:

The title should reflect the main finding, which is that KAP1 hp1 box somehow is important for its ability to silence elements.

'Constitutive' heterochromatin is an outdated concept, we know that heterochromatin at transposable elements can be dynamic / cell type specific and is not immutable as 'constitutive' implies.

Line 77, the comma is unneeded

Line 178, add 'we hypothesized' after 'therefore'

Line 171, 173, 180 - what is 601 DNA?

Line 190 - explain why a KAP1 fragment missing the first 110 amino acids was used, and refer to it as a fragment in every place its name is mentioned (instead of GFP-KAP1, say GFP-KAP1 fragment)

Line 290 - please mention KRAB zinc finger proteins as the mechanism of recruitment of KAP1 to specific DNA sequences.

Line 375 - typo, 'package'

Reviewer #3

(Remarks to the Author)

The work shows that Kap1 and HP1 co-occupy endogenous retroelements (ERVs) in mouse ESC and that the HP1 α chromoshadow domains bind the Kap1 Hbox. Crystal structures and NMR analyses combined with functional validation provide a high-resolution understanding of this interaction. Further biochemical characterization reveals that the Kap1 peptide enhances the HP1 CSD's interaction with DNA and its mobility in phase-separated droplets and heterochromatin in ESCs. When investigating the structure of the apo state of HP1 α CSDs the authors find a tetramer in the asymmetric unit with extra CSDs interacting with the β -sheet of the CSD asymmetrically and that the binding Kap1 resolves the asymmetry. Lastly, they show that the HP1 α -Kap1 interaction contributes to silencing the retroelements through epigenomic profiling of WT vs the V488E mutant.

This manuscript thoroughly analyses the HP1 α -Kap1 interaction and establishes its role in gene silencing ERVs in mice. The crystal structures and NMR analysis provide novel high-quality structural information. Furthermore, the work demonstrates that the interaction promotes the mobility of HP1 α -Kap1 within condensates of heterochromatin. Many of the results presented in this paper have in some form been studied before in the literature. Nevertheless, this is a comprehensive and high-quality study, which fills several important knowledge gaps and will be seen as a milestone in understanding the HP1 α -Kap1 silencing mechanism. In my opinion, this manuscript is suitable for publication in Nature Communications provided the issues raised below are addressed.

Issues:

1. Line 79 "suggests conflicting roles for HP1 proteins in maintaining ERVs in a transcriptionally repressed state [27-30]": I cannot see how these references support the argument for conflicting roles for HP1. This undermines the need for the research performed in this work.

2. The relevance of the tetrameric form observed in the apo structure or the helix observed in the crystal of the complex is unclear. These may be arrangements favoured by the crystal packing and functional evidence for its involvement in silencing should be provided. Otherwise, this must be qualified clearly as a structural observation of questionable importance.

Related to this, it should be clarified whether the peripheral CSDs observed in the apo state are engaged in dimerization

through crystal symmetry.

3. The mass photometry is inconclusive given the uncertainty in the masses measured. A spectrum of Kap1 alone is missing (Fig. 6f-h), which is required to interpret the mixture. Furthermore, mass photometry is likely performed under very low concentrations (concentrations are missing and should be specified in the figure legend or method). This will render the comparatively weak HP1-Kap1 complexes undetectable.

4. Ref 39 on line 249 seems to be out of place.

Reviewer #4

(Remarks to the Author)

In this paper, the authors investigated the mechanism of the HP1-KAP1 complex in silencing ERVs. Through structure-based methods, the authors identified the binding interface and the interacting details between HP1 and KAP1. They also found that KAP binding would enhance the DNA binding ability and facilitate HP1-mediated phase separation. Then through structure-based analyses, the authors found that the CSD of HP1 tends to oligomerize, and KAP1 binding would change the oligomerization state of the complex, which would promote the closed state of chromatin. In conclusion, through combined structural, biochemical and cell-biological methods, the authors provide us a mechanistic insight into the structure-function of HP1-KAP1 complex in heterochromatin formation at ERVs.

However, several issues may need to be addressed to clarify the conclusions the authors made.

1. As shown in Figure 4a, the difference in DNA-binding abilities between HP1 and its complex with the KAP1(hbox) does not seem very evident. As a chromatin-associated protein, KAP1 may also bind DNA. The KAP1(hbox) segment alone may not contribute much to DNA binding. The authors may use a large piece of KAP1, with its bound HP1, to do the EMSA analysis with DNA.

2. The authors did the assignment of the HP1 CSD domain alone first. Later using full length HP1, the authors found that most of the signals of the CSD domain were undetectable. I feel a little confused here. Does that mean the CSD domain alone and the same domain within the full length protein showed different signals? How do the authors explain the differences?

3. In the Supplementary Figure 6 caption, the authors titrated HP1 CSD domain with unlabeled HP1 CD domain, is there an error there? Why not titration with the CSD domain?

Version 1:

Reviewer comments:

Reviewer #1

(Remarks to the Author)

1- No evidence is provided in the revised version on the significance of the described tetramer of CSD. Gaurav et al., invoke previous results of Lechner et al. reported in reference 26. Lechner et al. were very cautious in interpreting their analytical ultracentrifugation data regarding the multimerization of this domain. On page 6457 they wrote the following: "The hCSD data were best described by a model containing dimers and tetramers (Fig. 7B). Although the apparent equilibrium constants returned by the fitting program for the dimer-tetramer association were quite similar (0.5 to 0.7 mM for four data sets), the data sets could not be fitted well with a single equilibrium constant." Gaurav et al. could be more cautious in interpreting their data. The crystal structure of CSD shows that it forms a homodimer. It also shows that each protomer interacts with a protomer of a neighboring dimer, and the protomers of this other dimer interact with protomers of neighboring dimers, and so on. Describing this arrangement as tetrameric is highly arbitrary (independently of the crystallographic symmetry) as there is an indefinite succession of dimers interacting with other dimers in the crystal lattice. The dimer-dimer interaction is very weak, as shown by the previous analytical centrifugation (apparent dissociation constant in the millimolar range) and the NMR data in Figure 5f (very small CSP values even using a 1:20 ratio). The system could be more adequately described as dimers in rapid equilibrium with very low populations of tetramers (and other higher-order oligomers; the higher the order, the lower the population). This would explain the broadening of NMR signals beyond detection (caused by exchange between a heterogeneous set of oligomers). The authors could have done a very simple experiment to verify their interpretation: a size exclusion chromatography with a multiangle light scattering detector on the wild-type protein and on a mutant designed to destabilize only the dimer-dimer interface observed in the crystal.

2- The suggestion was to look for duplication of signals in the HSQC of the complex, as expected by the formation of an asymmetric complex. If the authors had observed this duplication, the analysis would be more complete. If they cannot do it because the spectrum is not assigned, it is fine, the available crystal, fluorescence, and NMR data do support their interpretation.

3- No experimental evidence is provided in the revised version on the existence of tetramers of CSD when bound to the KAP1 peptide. I think that the conclusion that the binding of KAP1Hbox leads to a symmetric rearrangement of HP1 α CSD and multimerization is not supported by the data. I think that the arrangement of the molecules in the crystal lattice is overinterpreted, leading to a highly speculative multimerization model. Here too, the authors could also have done a very simple experiment to verify the formation of tetramers: a size exclusion chromatography with a multiangle light scattering detector on the complex with the wild-type protein or with a mutant designed to destabilize only the dimer-dimer interface.

4- No evidence is provided in the revised version supporting that KAP1 enhances liquid-liquid phase separation (LLPS). Figure 4c indicates that when KAP1 is present, the concentration of HP1a in the condensed phase increases relative to its concentration in the non-condense phase, but no changes in phase separation occur (unless this reviewer is missing something). The reference to the MOCHA-FRAP characterization of coacervates in reference 35 does not provide support for the conclusion that HP1a enhances LLPS in ESCs. Not only is the dip change very small, but the shape of the fluorescence recovery curves is more consistent with the existence of spatially clustered binding sites (ICBS) than with LLPs, as shown in reference 35. Perhaps experiments recorded over a longer recovery time and with smaller errors could clarify this issue.

5- The authors have improved the description of the methods in the revised version and clarified some issues. Perhaps the NMR section is excessive in detail. Still, some inconsistencies could be solved, like describing two different methods to quantify chemical shift perturbations (CSP and Δ combined) while in the figures only CSP is used, bar plots that are named "histograms", indicating only the name of the commercial kit (RED-NHS) for labeling HP1a in the methods, while the labeled protein is named HP1a-647N in results and figures. Some chemical descriptions could be improved: a hydroxyl does not donate a hydrogen bond (line 140), it donates a hydrogen atom to form a hydrogen bond.

Reviewer #2

(Remarks to the Author)

The authors addressed all my points in an appropriate manner and from my point of view this manuscript is now ready for publication.

Reviewer #3

(Remarks to the Author)

The authors have addressed most of my points and improved the manuscript significantly. However, in Issue 2, I asked for functional evidence to show the importance of multimerization in ERV silencing. The authors do not address this in the resubmission. Therefore, the biological role of the tetramer remains to be established. Given the other reviewer's comments, I see the manuscript as providing only an incremental advance over the published literature.

Reviewer #4

(Remarks to the Author)

The authors have addressed my queries.

Version 2:

Reviewer comments:

Reviewer #1

(Remarks to the Author)

The revised manuscript contains additional experiments and clarifications that have improved it. Some further clarifications are still necessary.

a. Figure 5c shows that HP1 α forms dimers at 10 nM concentration (top panel) and a mixture of trimers and higher-order oligomers at 40 nM (green panel), but mass photometry data in Figure 6h show that it forms a homogeneous population of trimers (74 kDa) at 50 nM. The possible formation of stable trimers and the discrepancy between Figures 5c and 6h should be discussed.

b. In line 262 it is stated that "Binding of the KAP1 Hbox short peptide further increased oligomerization of WT FL HP1 α in MP assays". However, the top panel of Figure 5j shows the formation of a homogenous complex with a 2:1 stoichiometry, and the measured mass is consistent with a tetramer. This should be stated and discussed.

c. The section heading in line 273 should include the words "in the crystal structure" at the end, as the experimental evidence presented corresponds to the crystal lattice.

d. The authors should explain why they think the oligomeric spiral ring observed in the crystal lattice of HP1 α -CSD bound to the KAP1 Hbox long peptide is biologically relevant. This is questionable in light of the heterogeneous mixtures of multimers observed with the full-length proteins in solution (Figure 6g and Supplemental Figure 12).

Minor comments:

- Please explain what "n-terminal extension" and "c-terminal extension" stand for in Figure 2. Human HP1 α spans residues 1-191. If non-native sequences are present at the chain ends, they should be described in the methods section, as they may influence oligomerization behavior.

- The triple-stranded beta-sheet described in line 131 is not represented as such in the ribbon diagram of Figure 2c (only two beta strands are indicated).
- Please use consistent labels in the mass photometry figures 5c, 5j, and 6h. Either use “apo FL HP1alfa” or “HP1alfa”, but the same in all of them.
- Please confirm that the dashed lines in Figure 5f indicate hydrogen bonds. The one between T130 and A148 connects the two carbonyls, which cannot form a hydrogen bond.
- Please confirm that the CSP values shown in Figure 3b are normalized ones. There is no normalization in equation number 2.

Reviewer #3

(Remarks to the Author)

The authors have addressed my concern and I deem the manuscript ready for publication.

We would like to thank the Reviewers for their insightful and very constructive comments, which were very helpful in revising and strengthening this manuscript (the responses are in blue).

Reviewer #1 (Remarks to the Author):

Gaurath et al. Report a structure-function study on the role of the HP1a interaction with KAP1 in silencing ERVs. While the functional studies appear to support such a role, the proposed mechanism, at least its structural aspect, is not clearly supported by the data.

Major concerns:

- The crystal structure of the apo form the CSD of HP1a shows a tetramer, and the NMR data is consistent with interactions other than the dimerization ones, as seen in the crystal. However, the affinity of this interaction is not quantified. Based on the magnitude of the CSP in Figure 5c this additional interaction seems to be very weak. The CSPs are about 10 times smaller than those caused by KAP1 peptide (even at lower molar ratios, figure 2c). As the interaction with the peptide is in the low microM range, the interaction with the -1 and +3 CSD protomers is, perhaps, in the milliM range, questioning the significance of the tetramer seen in the crystal.

The purpose of the experiment shown in Fig. 5f was to detect chemical shift changes occurred in residues other than the residues that mediate dimerization through α -helices, and because both CSD-CSD interactions, through α -helices and β -strands are present, we are unable to estimate binding affinities. We agree with the reviewer, the β -strands-mediated interaction is weaker than the α -helices-mediated interaction, but it is specific as the CD domain does not induce such changes (Suppl. Fig. 10). Please also see below, on page 2, a detailed response regarding the β -interfaces.

- Along the text, the symmetric nature of the complex between the CSD of HP1a and the peptide of KAP1 is stated and discussed. However, it seems to me that the complex is not symmetric. The dimeric structure of the CSD of HP1a in the apo form might be symmetric, but bound to the KAP1 peptide, as seen in the crystal structure, is asymmetric. The single KAP1 peptide as a beta strand parallel to one protomer and antiparallel to the other breaks the symmetry of the CSD dimer. NMR should be able to clarify this issue. The full HSQC spectrum of the CSD of HP1a in supp. Fig. 3 shows a single set of backbone amide signals, which is consistent with a symmetric oligomer, but the full spectrum of the CSD bound to the KAP1 peptide is not shown. If the complex is symmetric, a single set of signals should be observed. If it is asymmetric (as the crystal structure shows) two sets of signals should be observed, because of the magnetic inequivalence created by the single KAP1 peptide and the tight binding (in slow exchange). If only one set of signals is observed, it may be that the crystal structure does not adequately represent the complex in solution.

We appreciate this comment and have clarified description of symmetry. We refer exclusively to the symmetric arrangement of CSD (within the complex) but not to the entire complex with KAP1 peptide to avoid confusion. We have included new Suppl. Fig. 9 (also shown on the right to make it clearer). In the apo-state of HP1 α , the CSD1 and

CSD2 protomers are superimposed, but the peripheral CSD-1 and CSD' protomers cannot be superimposed. In contrast, in the complex with KAP1 peptide, CSD1 and CSD2 protomers are superimposed, as well as peripheral CSD-1 and CSD3 protomers are superimposed (with RMSDs = 0 Å), and there is a crystallographic 6-fold rotational symmetry between the dimer molecules.

as suggested, full HSQC titration has been added in Suppl. Fig. 3.

- Most of the panels in Figure 6, c are models built based on the interpretation of the crystal structures whose relevance must be reevaluated. Figure 6 is not a “diagram of the symmetrical tetramer of the HP1 α CSD-KAP1Hbox complex” as said in the figure legend, or at least experimental evidence of the existence of this tetramer has not been presented.
- Overall, I think that the conclusion that binding to KAP1a leads to a symmetric rearrangement of HP1a and multimerization is not sufficiently supported by the data.

We have revised Fig. 6a legend to: “A ribbon diagram of the tetramer of the HP1 α _{CSD}-KAP1_{Hbox}' complex.” As mentioned above, comparison of panels (b) and (c) in Suppl. Fig. 9 shows that CSDs of HP1 α undergo symmetric rearrangement upon binding of KAP1_{Hbox}'.

We have revised the title ‘Binding of KAP1...’ on page 10 to refer to CSD of HP1 α : Binding of KAP1_{Hbox}' leads to a symmetric rearrangement of HP1 α _{CSD}...

The existence of β -interfaces has been evaluated and confirmed by determining the crystal structure of the tetrameric CSD of HP1 α in complex with a peptide (green, not KAP1) designed to study the β -interfaces and their dynamics (please see Figure on the right, the manuscript is in preparation).

[REDACTED]

Regarding the oligomeric state: two experiments corroborated that HP1 α forms multimers larger than a dimer: the slow tumbling of CSD within full length HP1 α in NMR experiments and high molecular mass complexes observed in MP (please see Fig. 5a, 6g and Suppl. Fig. 12). Furthermore, our results support analytical ultracentrifugation studies of apo-HP1 α _{CSD} dimer-tetramer equilibrium (ref. ²⁶).

-The significance of some experiments is unclear. For instance, the assays on phase-separated samples shown in Figure 4c show that the concentration of HP1a in the droplets is higher when KAP1 is present than when absent (it recruits more HP1a to the droplets). This is consistent with KAP1 binding to DNA and HP1a. But what is the conclusion regarding the “phase separation ability of HP1a”? In the FRAP experiments, the dip is slightly higher when KAP1 is present, but it does not indicate that KAP1 enhances the ability of HP1a to create phase separation in ESCs.

We now refer to the original work (ref. ³⁵) that describes the MOCHA-FRAP for characterization of LLPS condensates in living cells and reports analysis and interpretation of the dip intensity. We have also revised the text on page 10 to clarify this.

- The experimental details in the methods section are insufficient to judge the experimental design and to reproduce the data. For instance, in mass photometry experiments, the volumes of samples are indicated, but not the relevant parameter, which is the protein concentration. This is just one example, not an exhaustive revision of the method details.

Proteins concentrations have been added to MP methods. NMR, protein production, EMSA and MP method sections have been expanded, and other methods are also updated.

- The labeling of some figures is cryptic or unclear. For instance, "DIC" in Figure 4c, the masses expressed in percentage in Figures 6f and h, "normalized CSP" in several figures (in equation 2 there is no normalization, the factor α is not given and the calculated quantity is not named CSP). I think that the authors should thoroughly re-evaluate the interpretation and presentation of the structural data as well as the description of the methods.

DIC, Differential interference contrast has been added to Fig. 4c legend. % of total counts in each peak in Fig. 6g, h has been removed and only masses in kDa remain. NMR method section has been revised to include the normalization formula, the factor α and the assignments protocol.

Reviewer #2 (Remarks to the Author):

I have read the manuscript titled "KAP1 organizes HP1 α for silencing of endogenous retroviral elements in embryonic stem cells". I am a researcher with more than a decade of experience working on TRIM28 and its role in epigenetic silencing of transposable elements. Their key result is the obtained structure of the KAP1 HP1 box peptide to a dimer of HP1, and some validation of established relationships between KAP1 and transposable elements. Here are a few comments that in my opinion would improve the text, as well as some concerns that I hope the authors will be able to address before further consideration for publication by the editor.

Major comments:

The authors need to tone down some claims, for example "Our findings provide mechanistic and functional insights essential in our understanding of how ERVs are silenced." could be "Our findings add some mechanistic and functional insights to our understanding of how ERVs are silenced."

This sentence has been revised to: Our findings provide mechanistic and functional insights that deepen our understanding of how ERVs are silenced.

Another example: "we demonstrate the critical role of HP1 α and KAP1 in silencing of repetitive elements" would be more appropriate to say "we confirm". Please review every claim related to mechanisms of ERV silencing, for which there is a large body of literature and a common understanding of the mechanism implicating H3K9me3 recruiting HP1 to form liquid-like phase separated heterochromatin domains.

As suggested, the word 'demonstrate' has been changed to 'confirm'.

Another example: "The findings that KAP1-containing complexes mediate transcriptional repression of ERVs and KAP1 binds to HP1 proteins prompted us to investigate whether KAP1 and HP1 cooperate in ERVs silencing." Rephrase as "prompted us to dig deeper in the mechanisms to explain how KAP1 and HP1 cooperate to mediate ERV silencing".

We have modified the text to incorporate this suggestion, clarifying the purpose of our study which is to investigate the role of the direct interaction between KAP1 and HP1 in ERV silencing.

Please rephrase the sentence at line 55 claiming "HP1 β and HP1 γ are implicated in both gene activation and repression, whereas HP1 α is associated with transcriptional silencing and chromosome segregation and is enriched at pericentric heterochromatin". The 'activation' claim is poorly supported, many of the references cited are correlative (for example, with transcription elongation and not 'activation') and outdated, and many of which are in organisms where KAP1 doesn't exist.

We agree and have replaced these two sentences to be more representative of our current understanding of HP1 function and heterochromatin formation, including updated reference to a 2023 review on heterochromatin formation by Shiv Grewal.

Line 111: "To better understand the high degree of co-localization of KAP1 with HP1s at ERVs, we explored the underlying mechanism." Please discuss the commonly accepted mechanism of colocalization, which is simply that KAP1 leads to H3K9me3 deposition, and HP1 binds H3K9me3.

We have updated the introduction of this section as suggested:

The commonly accepted mechanism for HP1s and KAP1 colocalization involves SETDB1 mediated deposition of H3K9me3 supported by KAP1 and recognized by HP1, and direct interaction between KAP1 and HP1 has also been reported^{6,19,20,26,29}.

Line 118: Rephrase "and a larger region of KAP1" to "and a larger KAP1 fragment around the HP1 box" – 'region' has been changed to 'fragment'

In that section and all remaining ones, when referring to this larger KAP1 HP1 box peptide, please refrain from using 'KAP1' to refer to it - switch every instance of KAP1 (including in figures) to something more appropriately describing the peptide. A clearer naming scheme which leaves no ambiguity when experiments are done with the larger KAP1 hp1 box fragment is important. Referring to it as 'KAP1' is too ambiguous and potentially misleading. – we agree and have labeled the larger KAP1 Hbox fragment as KAP1_{Hbox}. All fragment/peptide names in the text and labels in Figures have been updated.

Add a section discussing whether it is likely or not that the peptide would behave differently in the context of the full length protein (accessibility of various residues, surrounding charges, 3D conformation). Similarly, add a section discussing whether the KAP1 HP1 box V488E mutant could have possible interactions / effects on other domains of the full-length protein?

We discuss the effect of surrounding residues on page 9, and IP assays (Fig. 3g, h) indicate that this interaction occurs between the full-length proteins. The sentence on page 8 has been revised to: Together, these data pointed to the importance of hydrophobic and van der Waals contacts in stabilization of the HP1 α _{CSD}-KAP1_{Hbox} complex and that this complex is also formed in context of the full-length proteins.

"Unexpectedly, we found that while resonances of CD were observed in the 1H,15N HSQC spectrum of full-length HP1 α , most of the signals of HP1 α _{CSD} were undetectable, even when the spectra were collected on a 900 MHz NMR spectrometer (Fig. 5a, b). These data suggested that HP1 α _{CSD} may form oligomers larger than a dimer – HP1 α _{CSD} is a small globular module that should be amenable to NMR studies in the dimeric form – and corroborated analytical ultracentrifugation studies describing HP1 α _{CSD} dimer tetramer equilibrium." This section is arcane and non-descriptive to someone not familiar with NMR, please explain further in simpler terms.

We have added to the Fig. 5a legend: CSD signals in full length HP1 α are broadened beyond detection, indicating that the size of this part of HP1 α becomes too large, i.e. CSD multimerization (higher order than detectable dimerization), reduces tumbling rate.

Line 260 - ChIP-seq is notoriously non-quantitative, tone down conclusions that are weak based on quantification of partial complementation.

We are aware of and agree with the reviewer about difficulties in properly quantifying genomics experiments. We feel that the language in the manuscript is reflective of our ability to interpret the ATAC-seq data. For example, we state that exogenous expression of KAP1 is able to “partially restore” closed chromatin in KAP1 KO ESCs, that the HP1-KAP1 interaction “contributes to” closed chromatin, and that the V488E mutant could not restore closed chromatin “to the same extent”. We feel that this language is overall cautious in our interpretation of these data.

Line 268 - "Collectively, these results indicated that binding of KAP1 to HP1 has a role in silencing of ERVs in ESCs." This is known for decades, use 'confirm' instead of 'indicated'.

We have clarified that our study demonstrates that in addition to KAP1’s known role in HP1 recruitment through H3K9me3, the direct binding of KAP1 to HP1 has a role in silencing of ERVs in ESCs. This sentence on page 13 has been revised to:
Collectively, these results indicated that in addition to KAP1’s known role in HP1 recruitment through H3K9me3, the direct binding of KAP1 to HP1 has a role in silencing of ERVs in ESCs.

While discussing their model of HP1 / KAP1 interaction in link with the observed tetramer structure, have the authors considered a simple model where a dimer of KAP1, each recruiting a dimer of HP1, explains the stoichiometry of HP1 oligomerization in presence of KAP1? – we considered this, and MP data show that the complex (2 KAP1 + 4 HP1) of molecular mass of ~280 kDa is valid, however molecular masses of ~329 kDa and higher (see Suppl. Fig. 12) indicate the formation of larger complexes.

Figure 1g - please explain why there is no enrichment of any HP1 to sequences enriched in H3K9me3 and KAP1 - is the HP1 ChIP-seq less sensitive than others (signal-noise ratio)? Especially so that the signal is very much visible in the examples given in panel g? Is there some quantification mistake, for example counting all elements instead of just the bound ones?

The reviewer is correct in that we quantified all elements instead of just the bound ones for Figure 1g. As our goal was to show family-wide trends, we think that this is an appropriate representation of the data that should be maintained. The disconnect between the HP1 enrichment shown in the browser tracks in Figure 1h and the heat maps / average profiles shown in Figure 1i-k and the heat map shown Figure 1g is because the data was displayed as log2 of signal over input and did not take into account differences in signal-to-noise between data sets. Further, as the H3K9ac signal at LINEs is quite high, this further compressed the dynamic range of the color scale for other data sets. To better highlight trends between data sets, we now z-score normalize each data set to itself across all 1370 repetitive element families, provided as a replacement Supplemental Figure 1. From these 1370, we highlight 20 retroelements which show many of the overall trends within these data. This representation makes it clear that families which are enriched with H3K9me3 and KAP1 tend to also be enriched with HP1, in agreement with all other panels in the figure.

Figure 4a - can this band shift be explained by high amounts of HP1 simply clogging the gel, especially so that it is known to form phase-separated domains at high concentrations? The bands are very smeary which would support this and not binding of HP1 to the DNA. – we performed additional EMSAs (shown in new Suppl. Figs. 6 and 7) and obtained the same result, binding of KAP1_{Hbox} to CSD increases binding of HP1 α to DNA. Please note that the shadow is caused by reflection/refraction of light through the gel boundary at the bottom of each loading well. This shadow is observed in all wells, even in wells without proteins (lanes with the DNA:protein ratio of 1:0).

Figure 5c - is this 'tetramer' an artifact of crystallography (symmetry)? The outer 2 units mode of interaction is not particularly convincing - why are those outer units not dimerizing as well? – α -helices of the outer two CSDs are also involved in the α -helix-mediated dimerization. We have added to the Fig 5c legend: A protomer forming an α -helical contact with CSD-1 and a protomer forming an α -helical contact with CSD' are not shown for clarity.

Minor comments:

The title should reflect the main finding, which is that KAP1 hp1 box somehow is important for its ability to silence elements. – as suggested the title has been revised to: The HP1 box of KAP1 organizes HP1 α for silencing of endogenous retroviral elements in embryonic stem cells.

'Constitutive' heterochromatin is an outdated concept, we know that heterochromatin at transposable elements can be dynamic / cell type specific and is not immutable as 'constitutive' implies. – the word 'constitutive' has been removed

Line 77, the comma is unneeded – removed

Line 178, add 'we hypothesized' after 'therefore' – added

Line 171, 173, 180 - what is 601 DNA? – we have added '147 bp Widom 601 DNA' on page 8 and in Fig. 4a legend.

Line 190 - explain why a KAP1 fragment missing the first 110 amino acids was used, and refer to it as a fragment in every place its name is mentioned (instead of GFP-KAP1, say GFP-KAP1 fragment) – this construct of KAP1 was previously used (ref. 50), and as suggested, we refer to it as GFP-KAP1_{FR} fragment.

Line 290 - please mention KRAB zinc finger proteins as the mechanism of recruitment of KAP1 to specific DNA sequences. – we now mention that KRAB zinc finger proteins recruit KAP1 to specific DNA sequences and include new citations (ref. 43-45).

Line 375 - typo, 'package' – corrected

Reviewer #3 (Remarks to the Author):

The work shows that Kap1 and HP1 co-occupy endogenous retroelements (ERVs) in mouse ESC and that the HP1 α chromoshadow domains bind the Kap1 Hbox. Crystal structures and NMR analyses combined with functional validation provide a high-resolution understanding of this interaction. Further biochemical characterization reveals that the Kap1 peptide enhances the HP1 CSD's interaction with DNA and its mobility in phase-separated droplets and heterochromatin in ESCs. When investigating the structure of the apo state of HP1 α CSDs the authors find a tetramer in the asymmetric unit with extra CSDs interacting with the β -sheet of the CSD asymmetrically and that the binding Kap1 resolves the asymmetry. Lastly, they show that the HP1 α -Kap1 interaction contributes to silencing the retroelements through epigenomic profiling of WT vs the V488E mutant.

This manuscript thoroughly analyses the HP1 α -Kap1 interaction and establishes its role in gene silencing ERVs in mice. The crystal structures and NMR analysis provide novel high-quality structural information. Furthermore, the work demonstrates that the interaction promotes the mobility of HP1 α -Kap1 within condensates of heterochromatin. Many of the results presented in this paper have in some form been studied before in the literature. Nevertheless, this is a comprehensive and high-quality study, which fills several important knowledge gaps and will be seen as a milestone in understanding the HP1 α -Kap1 silencing mechanism. In my opinion, this manuscript is suitable for publication in Nature Communications provided the issues raised below are addressed.

Issues:

1. Line 79 "suggests conflicting roles for HP1 proteins in maintaining ERVs in a transcriptionally repressed state [27-30]": I cannot see how these references support the argument for conflicting roles

for HP1. This undermines the need for the research performed in this work. – this phrase has been removed

2. The relevance of the tetrameric form observed in the apo structure or the helix observed in the crystal of the complex is unclear. These may be arrangements favoured by the crystal packing and functional evidence for its involvement in silencing should be provided. Otherwise, this must be qualified clearly as a structural observation of questionable importance.

Related to this, it should be clarified whether the peripheral CSDs observed in the apo state are engaged in dimerization through crystal symmetry.

Formation of the multimers of apo-HP1 α larger than a dimer was confirmed by NMR (Fig. 5a). CSD signals in full length HP1 α are broadened beyond detection, indicating that the size of this part of HP1 α becomes too large, i.e. CSD multimerization (higher order than detectable dimerization), reduces tumbling rate. High molecular mass complexes observed in MP support the formation of HP1 α -KAP1 complexes of the order higher than a dimer (Fig. 6g and Suppl. Fig. 12). Our results also support previous analytical ultracentrifugation studies of apo-HP1 α _{CSD} dimer-tetramer equilibrium (ref. ²⁶).

We have clarified in the text (page 10) and in Fig. 5c legend that in the apo-HP1 α _{CSD} structure the peripheral CSD-1 is characterized by non-crystallographic symmetry and CSD' is characterized by crystallographic two-fold rotational symmetry. Spatial arrangement of CSDs in the unit cell of tetrameric apo-state of HP1 α is now shown in Suppl. Fig. 9.

3. The mass photometry is inconclusive given the uncertainty in the masses measured. A spectrum of Kap1 alone is missing (Fig. 6f-h), which is required to interpret the mixture. Furthermore, mass photometry is likely performed under very low concentrations (concentrations are missing and should be specified in the figure legend or method). This will render the comparatively weak HP1-Kap1 complexes undetectable. – the MS data for KAP1 have been added (Fig. 6i), and MP method section and figure legends have been updated.

4. Ref 39 on line 249 seems to be out of place. – this reference is correct; this publication contains the first reported ATAC-seq data sets from SETDB1 KO ESCs. An additional study reporting ATAC-seq data from SETDB1 KO cells, ref. 38, has also been added.

Reviewer #4 (Remarks to the Author):

In this paper, the authors investigated the mechanism of the HP1-KAP1 complex in silencing ERVs. Through structure-based methods, the authors identified the binding interface and the interacting details between HP1 and KAP1. They also found that KAP binding would enhance the DNA binding ability and facilitate HP1-mediated phase separation. Then through structure-based analyses, the authors found that the CSD of HP1 tends to oligomerize, and KAP1 binding would change the oligomerization state of the complex, which would promote the closed state of chromatin. In conclusion, through combined structural, biochemical and cell-biological methods, the authors provide us a mechanistic insight into the structure-function of HP1-KAP1 complex in heterochromatin

formation at ERVs.

However, several issues may need to be addressed to clarify the conclusions the authors made.

1. As shown in Figure 4a, the difference in DNA-binding abilities between HP1 and its complex with the KAP1(hbox) does not seem very evident. As a chromatin-associated protein, KAP1 may also bind DNA. The KAP1(hbox) segment alone may not contribute much to DNA binding. The authors may use a large piece of KAP1, with its bound HP1, to do the EMSA analysis with DNA.

Full length HP1 α has been shown to bind DNA through the hinge region linking CD and CSD, and here, we wanted to test if KAP1 binding to CSD affects the DNA binding activity of HP1 α . We have repeated EMSAs (shown in new Suppl. Fig. 6a) and obtained the same result, binding of KAP1 to CSD increases binding of full length HP1 α to DNA.

As suggested, we have also performed EMSA with the longer KAP1 construct (shown in new Suppl. Fig. 6b) and again obtained the same result, an increase in the DNA binding activity of HP1 α . As expected, the isolated HP1 α _{CSD} either in the presence or absence of KAP1_{Hbox} was unable to shift the DNA band (Suppl. Fig. 7a), supporting previous reports that the DNA binding activity of HP1 α depends on the hinge region^{13,32}. The KAP1 peptides without HP1 α did not appreciably bind DNA (Suppl. Fig. 7b), in agreement with previous reports showing that while KAP1 itself binds DNA, its HP1 box region is not involved^{33,34}. The text on pages 8-9 has been revised to include these data.

2. The authors did the assignment of the HP1 CSD domain alone first. Later using full length HP1, the authors found that most of the signals of the CSD domain were undetectable. I feel a little confused here. Does that mean the CSD domain alone and the same domain within the full length protein showed different signals? How do the authors explain the differences? – we have added to the Fig. 5a legend: CSD signals in full length HP1 α are broadened beyond detection, indicating that the size of this part of HP1 α becomes too large, i.e. CSD multimerization (higher order than detectable dimerization), reduces tumbling rate.

3. In the Supplementary Figure 6 caption, the authors titrated HP1 CSD domain with unlabeled HP1 CD domain, is there an error there? Why not titration with the CSD domain? – the CSPs induced by CSD are shown in Fig. 5f.

We would like to thank the Reviewers for their insightful and very constructive comments, which were very helpful in revising and strengthening this manuscript (the responses are in blue).

Reviewer #1 (Remarks to the Author) 1- No evidence is provided in the revised version on the significance of the described tetramer of CSD. Gaurav et al., invoke previous results of Lechner et al. reported in reference 26. Lechner et al. were very cautious in interpreting their analytical ultracentrifugation data regarding the multimerization of this domain. On page 6457 they wrote the following:
“The hCSD data were best described by a model containing dimers and tetramers (Fig. 7B). Although the apparent equilibrium constants returned by the fitting program for the dimer-tetramer association were quite similar (0.5 to 0.7 mM for four data sets), the data sets could not be fitted well with a single equilibrium constant.”

Gaurav et al. could be more cautious in interpreting their data. The crystal structure of CSD shows that it forms a homodimer. It also shows that each protomer interacts with a protomer of a neighboring dimer, and the protomers of this other dimer interact with protomers of neighboring dimers, and so on. Describing this arrangement as tetrameric is highly arbitrary (independently of the crystallographic symmetry) as there is an indefinite succession of dimers interacting with other dimers in the crystal lattice. The dimer-dimer interaction is very weak, as shown by the previous analytical centrifugation (apparent dissociation constant in the millimolar range) and the NMR data in Figure 5f (very small CSP values even using a 1:20 ratio). The system could be more adequately described as dimers in rapid equilibrium with very low populations of tetramers (and other higher-order oligomers; the higher the order, the lower the population). This would explain the broadening of NMR signals beyond detection (caused by exchange between a heterogeneous set of oligomers). The authors could have done a very simple experiment to verify their interpretation: a size exclusion chromatography with a multiangle light scattering detector on the wild-type protein and on a mutant designed to destabilize only the dimer-dimer interface observed in the crystal.

– we appreciate this comment and as suggested, have generated the mutant (L139E/L150E in the beta-interface, i.e. dimer-dimer interface) that disrupted only the association between dimers and tested this mutant. The term tetramer has been removed.

New data, shown in Fig. 5c, d, i, j, k demonstrate that HP1a oligomerization is mediated by the beta-interface of CSDs.

Fig. 5i- resonances of CSD in the HSQC spectrum of L139E/L150E FL HP1a became detectable, indicating that the formation of higher order oligomers is disrupted.

Fig. 5d- titration of WT CSD to the 15N-labeled I165E mutant of CSD [that cannot dimerize through the alpha-helices] leads to resonance perturbations, indicating that the mutant interacts with WT CSD but not through the alpha-helices.

Fig. 5c- Mass photometry data show that for WT FL HP1a dimer–dimer/higher order multimer equilibrium is concentration dependent. This is not observed for L139E/L150E FL HP1a, which remains largely dimeric.

Fig. 5k- L139E/L150E CSD is functional and binds KAP1.

Fig. 5j- Mass photometry data show that while binding of KAP1 peptide increases oligomerization of FL HP1a, the KAP1-promoted oligomerization is markedly decreased for the L139E/L150E mutant of HP1a.

We have revised the text on pages 11-12 to describe these new data and cite refs. 26, 37 that report that mutations in the beta-interface disrupt oligomerization. Ref. 37: “In solution, the HP1a-CSD molecule existed as a mixture of dimer and higher order oligomers that obviously decay the NMR signal. After a series of trial experiments with various mutants, we used the single mutant L139K [in the beta-interface] for the NMR study.”

2- The suggestion was to look for duplication of signals in the HSQC of the complex, as expected by the formation of an asymmetric complex. If the authors had observed this duplication, the analysis would be more complete. If they cannot do it because the spectrum is not assigned, it is fine, the available crystal, fluorescence, and NMR data do support their interpretation. – the spectrum of the complex is not assigned for this analysis.

3- No experimental evidence is provided in the revised version on the existence of tetramers of CSD when bound to the KAP1 peptide. I think that the conclusion that the binding of KAP1Hbox leads to a symmetric rearrangement of HP1 α CSD and multimerization is not supported by the data. I think that the arrangement of the molecules in the crystal lattice is overinterpreted, leading to a highly speculative multimerization model. Here too, the authors could also have done a very simple experiment to verify the formation of tetramers: a size exclusion chromatography with a multiangle light scattering detector on the complex with the wild-type protein or with a mutant designed to destabilize only the dimer-dimer interface.

– new mass photometry data (Fig. 5j and 5c) show that while binding of KAP1 peptide increases oligomerization of FL HP1a, the KAP1-promoted oligomerization is markedly decreased for the L139E/L150E mutant of HP1a. Oligomerization is further promoted by FL KAP1 (Fig. 6g and Suppl. Fig. 12).

4- No evidence is provided in the revised version supporting that KAP1 enhances liquid-liquid phase separation (LLPS). Figure 4c indicates that when KAP1 is present, the concentration of HP1a in the condensed phase increases relative to its concentration in the non-condense phase, but no changes in phase separation occur (unless this reviewer is missing something). The reference to the MOCHA-FRAP characterization of coacervates in reference 35 does not provide support for the conclusion that HP1a enhances LLPS in ESCs. Not only is the dip change very small, but the shape of the fluorescence recovery curves is more consistent with the existence of spatially clustered binding sites (ICBS) than with LLPs, as shown in reference 35. Perhaps experiments recorded over a longer recovery time and with smaller errors could clarify this issue.

– new data in KAP1^{-/-} ESCs stably expressing WT KAP1 or mutated KAP1 impaired in HP1 α binding show that the interaction with KAP1 enhances the formation of phase separated compartments of endogenous HP1 α in cells (Fig. 4c).

– to clarify FRAP assays in living cells, we have revised explanation of this experiment and analysis and also revised Fig. 4d to make it clearer that in the presence of GFP-KAP1, neither the bleached half of cellular heterochromatin nor the non-bleached half fully recovered their fluorescence and together with the non-bleach half showing dips at 1 s and 15 s indicate liquid-liquid phase separation properties.

5- The authors have improved the description of the methods in the revised version and clarified some issues. Perhaps the NMR section is excessive in detail. – we have shortened it
Still, some inconsistencies could be solved, like describing two different methods to quantify chemical shift perturbations (CSP and delta_combined) while in the figures only CSP is used, – made both consistent
bar plots that are named “histograms”, – changed to bar plots for NMR data
indicating only the name of the commercial kit (RED-NHS) for labeling HP1a in the methods, while the labeled protein is named HP1a-647N in results and figures – made consistent in the text and methods

Some chemical descriptions could be improved: a hydroxyl does not donate a hydrogen bond (line 140), it donates a hydrogen atom to form a hydrogen bond. – corrected

Reviewer #2 (Remarks to the Author)

The authors addressed all my points in an appropriate manner and from my point of view this manuscript is now ready for publication.

Reviewer #3 (Remarks to the Author)

The authors have addressed most of my points and improved the manuscript significantly. However, in Issue 2, I asked for functional evidence to show the importance of multimerization in ERV silencing. The authors do not address this in the resubmission. Therefore, the biological role of the tetramer remains to be established. Given the other reviewer's comments, I see the manuscript as providing only an incremental advance over the published literature.

Original comment:

2. The relevance of the tetrameric form observed in the apo structure or the helix observed in the crystal of the complex is unclear. These may be arrangements favoured by the crystal packing and functional evidence for its involvement in silencing should be provided. Otherwise, this must be qualified clearly as a structural observation of questionable importance. Related to this, it should be clarified whether the peripheral CSDs observed in the apo state are engaged in dimerization through crystal symmetry.

– we apologize for this omission on our part. We misunderstood the original comment (placed above for context) as a request for thorough analysis of the crystallographic packing and missed the point regarding evidence for the involvement of the beta-interface in silencing. We have removed the term tetramer and added data (new Fig. 5c, d, i, j, k) to show the importance of the beta-interface of CSDs HP1a in oligomerization.

We have obtained conditional HP1 KO ESCs (the genotype of these cells are *Cbx1*^{-/-};*Cbx3*^{fl/fl};*Cbx5*^{fl/fl};*Cre-ERT2*), treated these cells with 4-OHT to acutely deplete HP1 from ESCs and performed ATAC-seq, which shows that HP1 has a role in maintaining closed chromatin at retroelements in ESCs (new Suppl. Fig. 13). This had not been previously demonstrated. We next used these ESCs to develop a model in which WT HP1a, an alpha-interface I165E mutant, or a beta-interface L139E/L150E mutant could be expressed under the control of a doxycycline-inducible promoter with acute depletion of endogenous HP1. Unfortunately, we did not observe robust expression of the L139E/L150E mutant (Fig. on the left).

To avoid overinterpretation of ATAC-seq data obtained from this mutant cell line, we did not proceed with this experiment. However, we now refer to previous studies (ref. 26) in a different cell line reporting that mutations in the beta-interface show attenuated repressive function of HP1a in a luciferase assay, in support of the importance of this interface in HP1a-mediated silencing on page 12.

Fig. on the left shows residues mutated in ref. 26 that are mapped onto the HP1a CSD structure. Residues colored cyan (top panel) are primarily in the beta-interface. Mutations of these residues greatly decrease transcriptional repression function of HP1a-CSD. [L139 and L150 mutated in our study, see Fig. 5i, j, k, are colored red]. In contrast, mutations outside the beta-interface, alpha-interface, or KAP1-binding site (bottom panel, colored blue) do not affect the repressive activity of HP1a-CSD.

Reviewer #4 (Remarks to the Author)

The authors have addressed my queries.